# Phy-CoSF: Physics-Guided Continuous Spectral Fields Reconstruction and Spectral Super-Resolution for Snapshot Compressive Imaging

Wudi Chen [1]  Zhiyuan Zha [1]  Xin Yuan [2]  Shigang Wang [1]  Bihan Wen [3]  Jiantao Zhou [4]
Gang Yan [5]  Zipei Fan [6]  Ce Zhu [7]

## Abstract

Recent advances have demonstrated that coded aperture snapshot spectral imaging (CASSI) systems show great potential for capturing 3D hyperspectral images (HSIs) from a single 2D measurement. Despite the inherent spectral continuity of scenes captured by CASSI, most existing reconstruction methods are restricted to fixed, discrete spectral outputs, thereby precluding continuous spectral reconstruction or spectral super-resolution. To address this challenge, we propose Phy-CoSF, which synergizes deep unfolding networks with implicit neural representations, establishing a new paradigm for continuous spectral reconstruction and super-resolution in CASSI. Specifically, we propose a two-phase architecture that bridges discrete-wavelength training with continuous spectral rendering, enabling the synthesis of high-fidelity HSIs at arbitrary target wavelengths. At the core of our framework lies the continuous spectral fields (CoSF) module, embedded within each unfolding stage as a dynamic prior, which comprises a triple-branch cross-domain feature mixer for comprehensive spatial-frequency-channel feature fusion, alongside a spectral synthesis head that generates spectral intensities by querying continuous wavelength coordinates. Extensive experimental results demon-

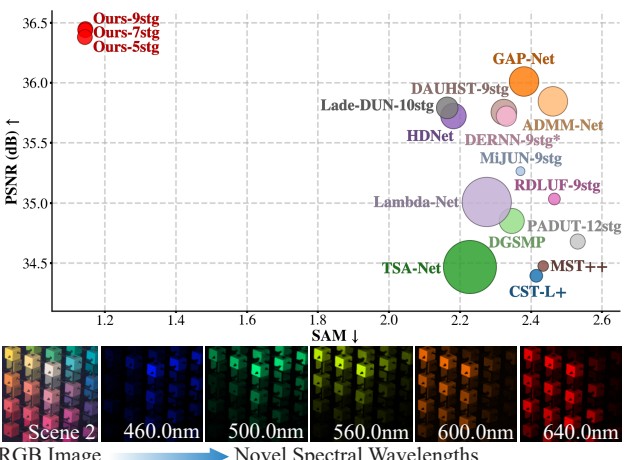

*Figure 1.* Comparison of continuous spectral reconstruction quality, spectral fidelity, and parameters (M). The bottom row visualizes the spectral super-resolution results at novel wavelengths.

strate that Phy-CoSF not only achieves continuous modeling at arbitrary spectral resolutions but also outperforms many state-of-the-art methods in both reconstruction fidelity and spectral detail preservation. Our code and more results are available at: https://github.com/PaiDii/Phy-CoSF.git.

## 1. Introduction

Coded aperture snapshot spectral imaging (CASSI) systems have demonstrated vast potential in medical diagnostics (Bjorgan & Randeberg, 2015; Meng et al., 2020c), precision agriculture (Lu et al., 2020), and remote sensing (Zhang et al., 2021; 2022) by enabling rapid 3D hyperspectral acquisition. However, retrieving high-fidelity HSIs from these compressed 2D measurements remains a severely ill-posed inverse problem. To tackle this challenge, reconstruction algorithms have evolved from traditional model-driven priors (*e.g.*, sparsity (Bioucas-Dias & Figueiredo, 2007) and low-rankness (Liu et al., 2018; Chen et al., 2023b; Luo et al., 2022)) to data-driven deep learning methodologies, including end-to-end (E2E) networks (Meng et al., 2020b; Huang et al., 2021; Cheng et al., 2022) and deep unfolding

[1]College of Communication Engineering, Jilin University, Changchun 130012, China. [2]School of Engineering, Westlake University, Hangzhou, Zhejiang 310024, China. [3]School of Electrical & Electronic Engineering, Nanyang Technological University, Singapore 639798. [4]Department of Computer and Information Science, University of Macau, Macau 999078, China. [5]College of Computer Science and Technology, Jilin University, Changchun 130012, China. [6]College of Artificial Intelligence, Jilin University, Changchun 130012, China. [7]School of Information and Communication Engineering, University of Electronic Science and Technology of China, Chengdu 611731, China. Correspondence to: Zhiyuan Zha <zhiyuan_zha@jlu.edu.cn>.

*Proceedings of the 43rd International Conference on Machine Learning*, Seoul, South Korea. PMLR 306, 2026. Copyright 2026 by the author(s).

networks (DUNs) (Wu et al., 2024; Dong et al., 2024; Qin et al., 2025). However, these methods are formulated on fixed, discrete spectral representations, which deviate from the intrinsic spectral continuity of the CASSI imaging principle and thereby preclude continuous spectral reconstruction or super-resolution.

Bearing the above concerns in mind, we propose Phy-CoSF, a novel physics-guided framework that synergizes the iterative optimization structure of DUNs with the modeling capabilities of INRs for continuous spectral reconstruction and super-resolution. Specifically, we establish a DUNs-based two-phase paradigm, empowering a model trained on discrete wavelengths to render at arbitrary continuous spectral coordinates, thereby realizing high-fidelity synthesis of target wavelengths and demonstrating superior continuous reconstruction performance, as illustrated in Figure 1. At the core of Phy-CoSF lies our proposed continuous spectral fields (CoSF) module, which replaces discrete priors to function as a dynamic continuous prior within each unfolding stage. It employs a triple-branch feature mixer to process multi-scale features at fine, meso, and coarse granularities, driven by a cross-domain feature encoder (CDFE) that fuses information across spatial, frequency, and channel domains. Within the frequency domain, we design a Fourier Mamba module to efficiently capture global dependencies. Building upon these comprehensive representations, our proposed spectral synthesis head (SSH) integrates them with high-frequency spectral embeddings to synthesize intensities at arbitrary spectral wavelengths, thereby transforming the discrete unfolding network into a physics-guided, continuous framework. To the best of our knowledge, this is the first work to jointly address continuous spectral reconstruction and super-resolution via a unified framework for CASSI. The main contributions of this paper are summarized as follows:

(1) We propose the Phy-CoSF framework for continuous spectral reconstruction and super-resolution, which incorporates implicit continuous modeling while preserving physical interpretability.

(2) We establish a train-render two-phase paradigm, which empowers models trained on discrete spectral bands to natively render arbitrary spectral resolutions.

(3) We design a CoSF module, achieving dynamic, continuous spectral modeling via a Fourier Mamba-based triple-branch cross-domain feature mixer and a SSH.

(4) Extensive experiments demonstrate that Phy-CoSF significantly outperforms many state-of-the-art methods in both reconstruction fidelity and spectral detail preservation.

**Conflict of Interest Disclosure.** The authors declare no conflict of interest.

## 2. Related Work

### 2.1. HSI Reconstruction

HSI reconstruction is inherently a severely ill-posed inverse problem. To address this problem, early methods mainly exploited model-driven priors, including sparsity (Bioucas-Dias & Figueiredo, 2007) and low-rankness (Liu et al., 2018; Chen et al., 2023b; Luo et al., 2022). In recent years, deep learning-based methods have shown promising performance in HSI reconstruction, which has been driven by their evolution from end-to-end (E2E) architectures (Meng et al., 2020b; Cheng et al., 2022; Huang et al., 2021; Chen et al., 2023a; Li et al., 2025) to physics-guided deep unfolding networks (DUNs) (Wu et al., 2024; Dong et al., 2024; Qin et al., 2025). E2E methods aim to directly learn the nonlinear mapping from the 2D measurement to the 3D HSI cube. For example, deep convolutional neural network (CNN)-based methods achieve promising reconstruction performance, including $\lambda$-Net (Miao et al., 2019), HDNet (Hu et al., 2022), BIRNAT (Cheng et al., 2022), and TSA-Net (Meng et al., 2020b). Despite achieving competitive performance, these methods suffer from limited interpretability. By prioritizing both model performance and interpretability, DUNs have become the mainstream framework for HSI reconstruction, which unrolls classical iterative algorithms (e.g., ADMM (Neal et al., 2011) and GAP (Liao et al., 2014)) into K-stage networks, such as ADMM-Net (Ma et al., 2019) and GAP-Net (Meng et al., 2020a). The core competitiveness of DUNs lies in the design of their prior modules, with many works explore a wide spectrum of powerful deep priors, from deep Gaussian scale mixture models (DGSMP) (Huang et al., 2021) to Transformer-based architectures such as MST++ (Cai et al., 2022b), CST-L+ (Cai et al., 2022a), and DAUHST (Cai et al., 2022c). Recently, by incorporating more complex strategies and priors, some works have achieved superior HSI reconstruction performance. For instance, DERNN-LNLT (Dong et al., 2024) introduced an efficient local-non-local Transformer alongside RNN-style cross-stage parameter sharing for model compression. LADE-DUN (Wu et al., 2024) embedded a pre-trained latent diffusion model as a generative prior. MiJUN (Qin et al., 2025) combined the Mamba architecture with tensor mode-k unfolding for efficient long-range dependency modeling. However, these methods are constrained by their inherently discrete spectral architectures, which prevents them from simultaneously achieving continuous spectral reconstruction and arbitrary spectral super-resolution. In contrast, our proposed approach establishes an entirely new continuous reconstruction paradigm, demonstrating significant superiority in both reconstruction fidelity and spectral

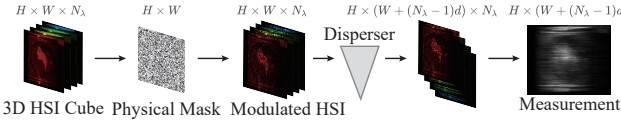

3D HSI Cube  Physical Mask  Modulated HSI          Measurement

*Figure 2.* Schematic of the CASSI system.

detail preservation.

## 3. Methodology

### 3.1. Problem Formulation and Physics-Guided Reconstruction Framework

**CASSI System.** CASSI (Arce et al., 2013) is a compressed sensing architecture that compresses a 3D HSI cube $\boldsymbol{X} \in \mathbb{R}^{H \times W \times N_\lambda}$ into a single 2D measurement $\boldsymbol{Y} \in \mathbb{R}^{H \times \tilde{W}}$, where $H$ and $W$ denote the spatial dimensions, $N_\lambda$ is the number of spectral channels, $\tilde{W} = W + (N_\lambda - 1)d$, and $d$ represents the dispersion shift per channel. As illustrated in Figure 2, the physical process first involves spatial modulation via a physical mask $\boldsymbol{M}$, followed by wavelength-dependent spatial shifts induced by a disperser, and finally, integration by the 2D sensor to form the snapshot:

$$\boldsymbol{Y} = \sum_{n_\lambda=1}^{N_\lambda} \tilde{\boldsymbol{X}}(:,:,n_\lambda) \odot \tilde{\boldsymbol{M}}(:,:,n_\lambda) + \boldsymbol{B}, \quad (1)$$

where $\tilde{\boldsymbol{X}} \in \mathbb{R}^{H \times \tilde{W} \times N_\lambda}$ and $\tilde{\boldsymbol{M}} \in \mathbb{R}^{H \times \tilde{W} \times N_\lambda}$ denote the spatially dispersed versions of the HSI and the mask, respectively, $\odot$ denotes element-wise multiplication, and $\boldsymbol{B} \in \mathbb{R}^{H \times \tilde{W}}$ represents additive sensor noise.

This forward imaging process can be simplified into a linear degradation model:

$$\boldsymbol{y} = \boldsymbol{\Phi}\boldsymbol{x} + \boldsymbol{n}, \quad (2)$$

where $\boldsymbol{x} \in \mathbb{R}^{H\tilde{W}N_\lambda}$ and $\boldsymbol{y} \in \mathbb{R}^{H\tilde{W}}$ denote the vectorized HSI and measurement, respectively, $\boldsymbol{n}$ is noise, and $\boldsymbol{\Phi} \in \mathbb{R}^{H\tilde{W} \times H\tilde{W}N_\lambda}$ is the sensing matrix encapsulating the modulation and dispersion operations. Since $H\tilde{W} \ll H\tilde{W}N_\lambda$, recovering $\boldsymbol{x}$ from $\boldsymbol{y}$ is a severely ill-posed inverse problem.

**Physics-Guided Iterative Unfolding Framework.** To solve this ill-posed inverse problem, it is commonly formulated as the following minimization problem, comprising a data fidelity term and a regularization prior term:

$$\hat{\boldsymbol{x}} = \arg\min_{\boldsymbol{x}} \frac{1}{2}\|\boldsymbol{y} - \boldsymbol{\Phi}\boldsymbol{x}\|^2 + \tau R(\boldsymbol{x}), \quad (3)$$

where $R(\boldsymbol{x})$ denotes the prior imposed on $\boldsymbol{x}$, and $\tau$ is the regularization parameter.

We employ a deep unfolding network (DUN) framework, grounded in the acceleration strategy-based half-quadratic

splitting (A-HQS) (Qin et al., 2025) algorithm, to solve this problem. By introducing an auxiliary variable $\boldsymbol{z} \in \mathbb{R}^{H\tilde{W}N_\lambda}$, this optimization problem is decomposed into $K$ stages of alternating iterations. At the $k$-th stage, the following steps are performed:

$$\boldsymbol{x}_{k+1} = \arg\min_{\boldsymbol{x}} \frac{1}{2}\|\boldsymbol{y} - \boldsymbol{\Phi}\boldsymbol{x}\|^2 + \frac{\mu}{2}\|\boldsymbol{x} - \hat{\boldsymbol{z}}_k\|^2, \quad (4)$$

$$\boldsymbol{z}_{k+1} = \arg\min_{\boldsymbol{z}} \frac{\mu}{2}\|\boldsymbol{z} - \boldsymbol{x}_{k+1}\|^2 + \tau R(\boldsymbol{z}), \quad (5)$$

$$\hat{\boldsymbol{z}}_{k+1} = \boldsymbol{z}_{k+1} + \beta_k(\boldsymbol{z}_{k+1} - \boldsymbol{z}_k), \quad (6)$$

where $\mu$ is the penalty parameter and $\beta_k$ denotes the momentum parameter at the $k$-th stage. Eq. (4) is the data fidelity step, which admits a closed-form solution: $\boldsymbol{x}_{k+1} = (\boldsymbol{\Phi}^T\boldsymbol{\Phi} + \mu\boldsymbol{I})^{-1}(\boldsymbol{\Phi}^T\boldsymbol{y} + \mu\hat{\boldsymbol{z}}_k)$. Eq. (5) and Eq. (6) are the prior regularization step and the acceleration step, respectively.

### 3.2. Overall Architecture

Grounded in the A-HQS iterative framework, we propose the Phy-CoSF network. As illustrated in Figure 3(a), Phy-CoSF unrolls a K-stage iterative process into a deep network. Within each stage, we explicitly instantiate the three core update steps of the A-HQS algorithm. First, we employ a degradation-aware network (DAN) (Dong et al., 2023) (shown in Figure 3(b)) to solve the data subproblem, which takes the accelerated prior estimate $\hat{\boldsymbol{z}}_{k-1}$ from the previous stage as input. The DAN module explicitly incorporates the physical mask $\boldsymbol{\Phi}$ and its transpose $\boldsymbol{\Phi}^T$ within its computational graph, thereby enforcing the data fidelity constraint during the update process to produce the data-fidelity estimate $\boldsymbol{x}_k$. Second, we propose a CoSF module as a learnable prior operator to solve the prior subproblem. This module takes $\boldsymbol{x}_k$ and a learnable noise level $\eta = \sqrt{\tau/\mu}$ as input and outputs the regularized estimate $\boldsymbol{z}_k$, i.e., $\boldsymbol{z}_k = \text{CoSF}(\boldsymbol{x}_k, \eta)$. Finally, $\boldsymbol{z}_k$ is updated according to Eq. (6) to obtain the accelerated estimate $\hat{\boldsymbol{z}}_k$, which serves as the input for the next stage.

**The Train-Render Two-Phase Paradigm.** To enable novel wavelength synthesis while optimizing on discrete data, we devise a train-render two-phase paradigm that decouples the discrete optimization process from the continuous rendering capability. During the training phase, the network is optimized on a fixed quantity of randomly sampled discrete target wavelengths. Specifically, the CoSF module only queries these ground-truth-corresponding wavelength coordinates, and the network is optimized by minimizing the L1 reconstruction loss $\mathcal{L}_{rec}$ on the associated spatial slices. During the rendering phase, the continuous modeling capability of Phy-CoSF is unleashed. Leveraging the continuous function map learned by the CoSF module, we can query arbitrary continuous wavelength coordinates. The

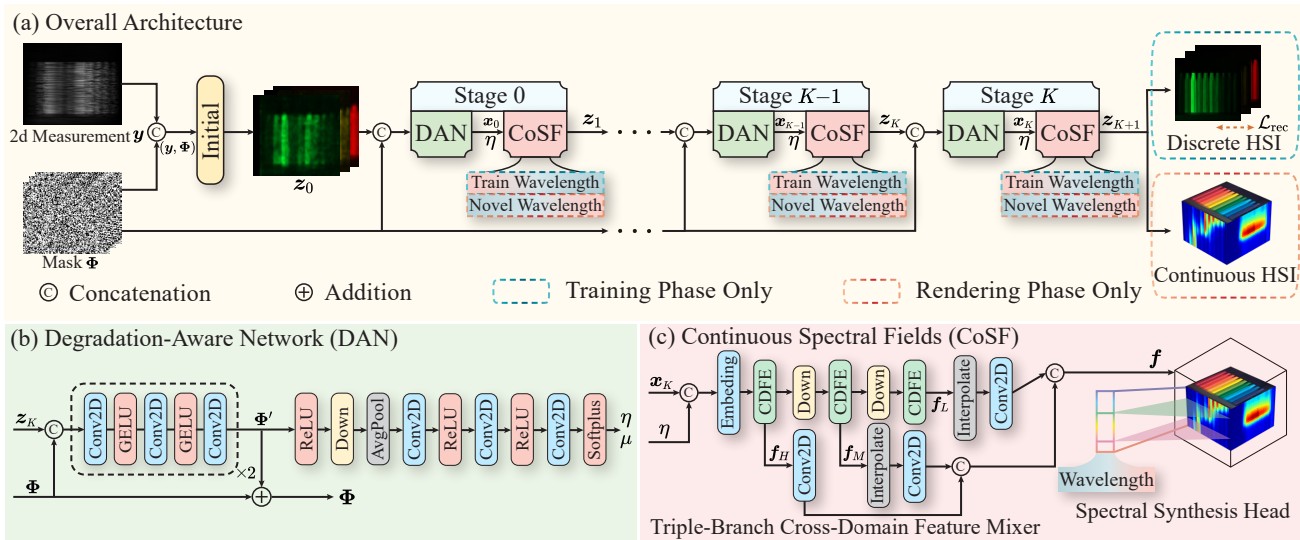

*Figure 3.* (a) The overall architecture of the proposed Phy-CoSF. The network employs a $K$-stage iterative framework, taking the 2D measurement $\boldsymbol{y}$ and the physical mask $\boldsymbol{\Phi}$ as input. The framework operates in two distinct phases: a training phase that utilizes discrete training wavelengths and a rendering phase capable of generating outputs at arbitrary continuous wavelengths. (b) In each stage, the DAN module explicitly incorporates $\boldsymbol{\Phi}$ to process the prior-regularized feature $\boldsymbol{z}_k$ into the data-fidelity estimate $\boldsymbol{x}_k$. (c) The CoSF module then takes $\boldsymbol{x}_k$ and a learnable noise level $\eta$ as input to produce the updated prior feature $\boldsymbol{z}_{k+1}$ through a triple-branch cross-domain feature mixer and a spectral synthesis head, enabling continuous spectral reconstruction and arbitrary spectral super-resolution.

network will natively render high-fidelity spectral intensities for these new coordinates, thus achieving arbitrary spectral super-resolution in a zero-shot manner.

### 3.3. Continuous Spectral Fields Prior Module

To overcome the inherent discrete limitations of traditional priors used in deep unfolding networks, we design the CoSF module, as shown in Figure 3(c). The CoSF module comprises a triple-branch cross-domain feature mixer and a spectral synthesis head (SSH). The core principle of this module is to enable continuous spectral synthesis by dynamically combining a wavelength-agnostic content representation with arbitrary continuous spectral coordinates. Specifically, the feature mixer first extracts a comprehensive latent representation $\boldsymbol{f} \in \mathbb{R}^{C \times H \times W}$ from $\boldsymbol{x}_k$ and the noise level $\eta$, where $C$ denotes the channel dimension. This representation $\boldsymbol{f}$ exclusively encodes the spatial structure, texture, and contextual information of the scene. Subsequently, the SSH aggregates $\boldsymbol{f}$ with the queried continuous wavelength coordinates to synthesize the corresponding spectral intensity at any specified coordinate.

**Triple-Branch Cross-Domain Feature Mixer.** HSIs intrinsically exhibit complex spatial structures and spectral correlations, with features distributed across multiple scales and domains. To capture this hierarchical information comprehensively, we propose a tri-level multi-scale architecture that processes features at fine, meso, and coarse granularities.

First, the input of the current stage $\boldsymbol{x}_k$ is concatenated with a $\eta$ and processed by a $3 \times 3$ convolutional embedding layer. This layer performs shallow feature extraction and channel upscaling. The resulting tensor is immediately fed into the first CDFE to generate the fine-grained feature $\boldsymbol{f}_H \in \mathbb{R}^{\frac{C}{12} \times H \times W}$, which strictly preserves spatial textures and local spectral details at full resolution. Subsequently, to capture broader contexts at meso- and coarse- granularities, the $\boldsymbol{f}_H$ is successively downsampled via a $4 \times 4$ convolution and processed by the subsequent CDFEs to generate the meso-scale feature $\boldsymbol{f}_M \in \mathbb{R}^{\frac{C}{6} \times \frac{H}{2} \times \frac{W}{2}}$ and the coarse-scale feature $\boldsymbol{f}_L \in \mathbb{R}^{\frac{C}{3} \times \frac{H}{4} \times \frac{W}{4}}$ in a cascaded manner, as expressed by:

$$\boldsymbol{f}_H = \text{CDFE}_H(Embed(\boldsymbol{x}_k, \eta)), \tag{7}$$

$$\boldsymbol{f}_M = \text{CDFE}_M(Down(\boldsymbol{f}_H)), \tag{8}$$

$$\boldsymbol{f}_L = \text{CDFE}_L(Down(\boldsymbol{f}_M)). \tag{9}$$

To effectively fuse the multi-granularity features extracted from the triple-branch architecture, we first harmonize their heterogeneous spatial scales. Specifically, the meso-scale feature $\boldsymbol{f}_M$ and the coarse-scale feature $\boldsymbol{f}_L$ are restored to the original high resolution via interpolation operations. Subsequently, to ensure channel consistency and mitigate potential aliasing artifacts induced by upsampling, the original high-resolution feature and the two upsampled streams are processed by distinct $1 \times 1$ refinement convolutions. Finally, these spatially aligned representations are aggregated via channel-wise concatenation to produce the final

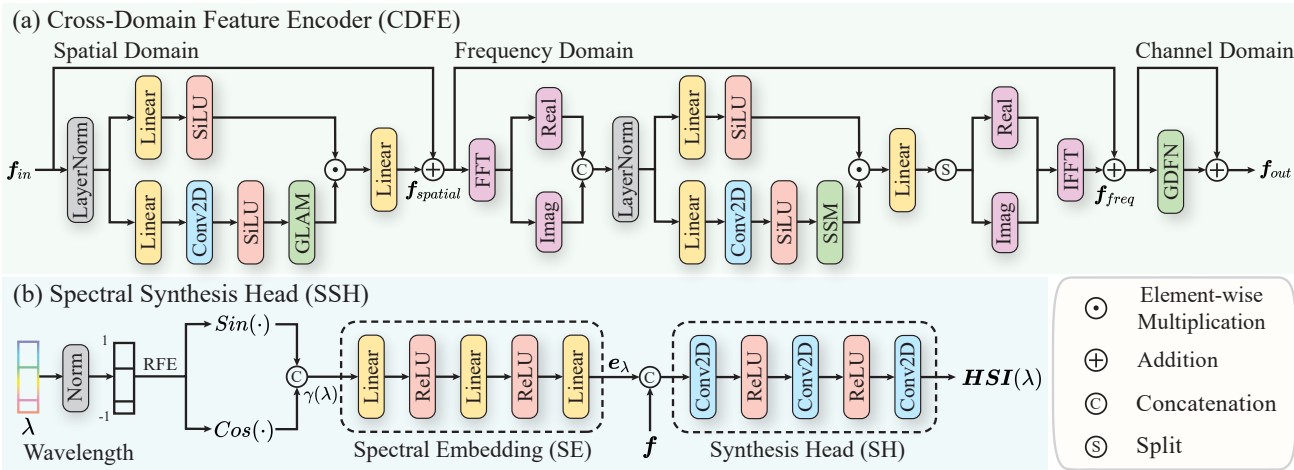

*Figure 4.* (a) The architecture of the CDFE module, which fuses information from the spatial, frequency, and channel domains to produce a cross-domain, wavelength-agnostic feature $\boldsymbol{f}_{out}$. (b) The structure of the SSH module. It encodes a continuous wavelength coordinate using random frequency encoding ($Sin(\cdot)$ and $Cos(\cdot)$) to form a spectral embedding, which is then combined with the feature $\boldsymbol{f}$ and processed by a synthesis head to generate the spectral image at the queried wavelength.

wavelength-agnostic latent representation $\boldsymbol{f}$, formulated as:

$$\boldsymbol{f} = \text{Concat}(Conv_H(\boldsymbol{f}_H), Conv_M(Interp_M(\boldsymbol{f}_M)),$$
$$Conv_L(Interp_L(\boldsymbol{f}_L))). \quad (10)$$

**Cross-Domain Feature Encoder.** The cross-domain feature encoder is designed to extract and fuse complementary prior information from the input features at each scale of the mixer. HSIs simultaneously contain local textures in the spatial domain, spectral correlations in the channel domain, and global structures in the frequency domain. To comprehensively capture such information, the CDFE adopts a serial backbone design, processing features sequentially through the spatial, frequency, and channel domains, as illustrated in Figure 4(a). A key property of this serial design is that the feature dimensionality is preserved across all domains. The spatial-domain module introduces the global-local attention mechanism (GLAM) network (Qin et al., 2025), whose core function is to process the input feature $\boldsymbol{f}_{in}$ to extract and refine local spatial structures and texture information. This module outputs the spatial-domain feature $\boldsymbol{f}_{spatial}$, which emphasizes local context and high-frequency spatial details, formulated as:

$$\boldsymbol{f}_{spatial} = \boldsymbol{f}_{in} + \text{GLAM Net}(\boldsymbol{f}_{in}). \quad (11)$$

To overcome the inherent limitations of convolution in capturing global context and long-range dependencies, we propose a frequency-domain Mamba module. This spatial-frequency transformation strategy effectively aggregates local spatial structures and global frequency dependencies to generate a comprehensive image representation. Specifically, we first map $\boldsymbol{f}_{spatial}$ to the frequency domain via the 2D fast Fourier transform (FFT), transforming spatial

information into a global and compact representation in which each frequency coefficient intrinsically encapsulates structural information from the entire spatial plane. Subsequently, the frequency-domain feature map is flattened into a 1D sequence and fed into a Mamba block (Zhu et al., 2024) for efficient long-range dependency modeling, capturing interrelationships among different frequency components. Finally, the refined sequence is reshaped and transformed back to the spatial domain via the 2D inverse fast Fourier transform (iFFT), and combined with the original spatial feature through a residual connection to yield $\boldsymbol{f}_{freq}$:

$$\boldsymbol{f}_{freq} = \boldsymbol{f}_{spatial} + iFFT(\text{Mamba}(FFT(\boldsymbol{f}_{spatial}))). \quad (12)$$

The feature $\boldsymbol{f}_{freq}$, having been jointly refined in the spatial and frequency domains, is fed into the channel-domain GDFN module (Dong et al., 2024). This module is designed to adaptively recalibrate feature responses across different channels, dynamically enhancing salient spectral features while suppressing potential noise. This process yields the refined feature $\boldsymbol{f}_{out}$:

$$\boldsymbol{f}_{out} = \boldsymbol{f}_{freq} + \text{GDFN}(\boldsymbol{f}_{freq}). \quad (13)$$

**Spectral Synthesis Head.** The spectral synthesis head is a key component of our framework for achieving continuous spectral reconstruction. Its function is to convert the wavelength-agnostic latent representation $\boldsymbol{f}$ into a high-fidelity, wavelength-specific spectral image. This process involves two core steps: continuous wavelength coordinate encoding and spectral intensity synthesis. Due to the tendency of deep networks to preferentially learn low-frequency functions, it is challenging to directly fit the high-frequency, fine-grained variations commonly observed in hyperspectral signals. Therefore, we adopt a strategy that combines a fixed

*Table 1.* Quantitative evaluation of continuous spectral reconstruction across ten scenes from ICVL dataset and the overall average. Metrics reported are SAM, PSNR (dB), and SSIM. '-9stg' indicates 9 unfolding stages. Best results are highlighted in **bold**.

| Algorithms | Params(M) | FLOPs(G) | Scene1 | Scene2 | Scene3 | Scene4 | Scene5 | Scene6 | Scene7 | Scene8 | Scene9 | Scene10 | Avg |
|---|---|---|---|---|---|---|---|---|---|---|---|---|---|
| MST++ (Cai et al., 2022b) | 0.07 | 1.18 | 2.60 | 3.70 | 2.17 | 2.08 | 1.68 | 1.75 | 2.36 | 2.26 | 2.29 | 3.46 | 2.43 |
|  |  |  | 30.91 | 29.99 | 35.96 | 39.60 | 39.39 | 40.09 | 37.53 | 32.60 | 27.44 | 31.24 | 34.48 |
|  |  |  | 0.820 | 0.798 | 0.939 | 0.950 | 0.947 | 0.944 | 0.922 | 0.879 | 0.765 | 0.878 | 0.884 |
| CST-L+ (Cai et al., 2022a) | 0.15 | 3.94 | 2.46 | 3.54 | 2.14 | 2.14 | 1.82 | 1.90 | 2.31 | 2.26 | 2.18 | 3.41 | 2.41 |
|  |  |  | 30.77 | 29.91 | 35.76 | 39.45 | 39.34 | 40.14 | 37.45 | 32.48 | 27.31 | 31.35 | 34.39 |
|  |  |  | 0.813 | 0.795 | 0.937 | 0.948 | 0.947 | 0.944 | 0.920 | 0.876 | 0.760 | 0.881 | 0.882 |
| GAP-Net (Meng et al., 2020a) | 4.21 | 65.73 | 2.65 | 3.62 | 1.97 | 1.92 | 1.61 | 1.73 | 2.25 | 2.20 | 2.42 | 3.43 | 2.38 |
|  |  |  | **32.42** | 31.29 | 38.79 | 41.57 | 40.84 | 40.99 | 39.63 | 33.70 | 28.38 | 32.51 | 36.01 |
|  |  |  | **0.871** | 0.852 | 0.964 | 0.969 | 0.960 | 0.954 | 0.949 | 0.906 | **0.812** | 0.908 | 0.915 |
| DAUHST-9stg (Cai et al., 2022c) | 2.42 | 6.68 | 2.51 | 3.43 | 1.96 | 1.83 | 1.61 | 1.69 | 2.20 | 2.13 | 2.40 | 3.47 | 2.32 |
|  |  |  | 32.26 | 31.06 | 38.25 | 41.17 | 40.40 | 40.76 | 39.39 | 33.74 | 28.24 | 32.30 | 35.76 |
|  |  |  | 0.867 | 0.845 | 0.961 | 0.966 | 0.958 | 0.952 | 0.946 | 0.905 | 0.810 | 0.903 | 0.911 |
| PADUT-12stg (Li et al., 2023) | 0.29 | 4.64 | 2.76 | 3.96 | 2.24 | 2.08 | 1.70 | 1.77 | 2.46 | 2.31 | 2.48 | 3.56 | 2.53 |
|  |  |  | 31.13 | 30.09 | 35.92 | 39.62 | 39.71 | 40.53 | 37.60 | 33.00 | 27.79 | 31.41 | 34.68 |
|  |  |  | 0.834 | 0.808 | 0.943 | 0.953 | 0.952 | 0.950 | 0.927 | 0.889 | 0.790 | 0.886 | 0.893 |
| RDLUF-MixS$^2$-9stg (Dong et al., 2023) | 0.11 | 31.49 | 2.71 | 3.80 | 2.16 | 2.00 | 1.66 | 1.77 | 2.32 | 2.29 | 2.48 | 3.49 | 2.47 |
|  |  |  | 31.44 | 30.30 | 36.75 | 40.21 | 40.03 | 40.66 | 38.31 | 33.13 | 27.80 | 31.70 | 35.03 |
|  |  |  | 0.845 | 0.819 | 0.950 | 0.958 | 0.955 | 0.951 | 0.935 | 0.895 | 0.795 | 0.892 | 0.900 |
| DERNN-LNLT*-9stg (Dong et al., 2024) | 0.93 | 122.14 | 2.58 | 3.42 | 1.94 | 1.87 | 1.59 | 1.70 | 2.22 | 2.13 | 2.40 | 3.47 | 2.33 |
|  |  |  | 32.05 | 31.02 | 38.47 | 41.47 | 40.45 | 41.07 | 39.32 | 33.64 | 27.69 | 32.05 | 35.72 |
|  |  |  | 0.865 | 0.848 | 0.962 | 0.968 | 0.959 | 0.955 | 0.946 | 0.904 | 0.803 | 0.900 | 0.911 |
| Lade-DUN-10stg (Wu et al., 2024) | 1.23 | 8.34 | 2.24 | 3.15 | 1.85 | 1.68 | 1.46 | 1.60 | 2.09 | 2.04 | 2.24 | 3.29 | 2.16 |
|  |  |  | 32.29 | 31.19 | 38.33 | **41.87** | 40.64 | 40.91 | 39.30 | 33.65 | 27.40 | 32.32 | 35.79 |
|  |  |  | 0.872 | 0.854 | 0.963 | **0.970** | 0.960 | 0.955 | 0.948 | **0.907** | 0.808 | 0.907 | 0.914 |
| Mijun-9stg (Qin et al., 2025) | 0.04 | 6.01 | 2.60 | 3.62 | 1.93 | 1.88 | 1.61 | 1.71 | 2.20 | 2.16 | 2.47 | 3.52 | 2.37 |
|  |  |  | 31.58 | 30.42 | 37.66 | 40.41 | 40.20 | 40.63 | 39.09 | 33.27 | 27.72 | 31.66 | 35.26 |
|  |  |  | 0.849 | 0.825 | 0.957 | 0.960 | 0.956 | 0.951 | 0.943 | 0.896 | 0.786 | 0.890 | 0.901 |
| **Phy-CoSF-9stg** | 0.27 | 801.38 | **1.39** | **1.70** | **0.82** | **1.12** | **0.97** | **1.08** | **1.00** | **0.90** | **1.15** | **1.26** | **1.14** |
|  |  |  | 32.25 | **31.76** | **40.21** | 41.66 | **40.92** | **41.25** | **40.09** | **34.32** | **28.55** | **33.47** | **36.45** |
|  |  |  | 0.865 | **0.857** | **0.967** | 0.967 | **0.961** | **0.955** | **0.949** | 0.905 | 0.805 | **0.916** | **0.915** |

high-frequency inductive bias with a learnable non-linear mapping to encode the continuous scalar wavelength coordinate $\lambda \in \mathbb{R}^1$ into an information-rich, high-dimensional feature representation. Specifically, we first normalize $\lambda$ to the range $[-1, 1]$, and then feed it into a fixed random frequency encoder (RFE), which utilizes a set of frequency vectors $\boldsymbol{b}$ randomly sampled from a Gaussian distribution $\mathcal{N}(0, \sigma^2)$ and kept fixed during training, defined as:

$$\gamma(\lambda) = \Big[ \sin(2\pi\lambda\boldsymbol{b}_1), \dots, \sin(2\pi\lambda\boldsymbol{b}_m), \\ \cos(2\pi\lambda\boldsymbol{b}_1), \dots, \cos(2\pi\lambda\boldsymbol{b}_m) \Big], \tag{14}$$

where $m$ denotes the mapping dimension, and $\sigma$ is the standard deviation.

This continuous wavelength coordinate encoding step provides a powerful inductive bias for the network by explicitly mapping the 1D coordinate into a high-dimensional space spanned by random Fourier features. However, fixed basis functions alone lack the flexibility to adapt to specific spectral signatures. Therefore, we further pass $\gamma(\lambda)$ into a learnable spectral embedding (SE) module, implemented as a lightweight MLP, to project it into a more expressive, task-relevant latent embedding space $\boldsymbol{e}_\lambda \in \mathbb{R}^D$, as:

$$\boldsymbol{e}_\lambda = \text{SE}(\gamma(\lambda)), \tag{15}$$

where $D$ denotes the latent embedding dimension.

In the spectral intensity synthesis step, $\boldsymbol{e}_\lambda$ is concatenated with the wavelength-agnostic latent representation $\boldsymbol{f}$ to form a fused feature tensor that incorporates both spatial content and spectral coordinate-specific information. This tensor is then fed into a lightweight convolutional synthesis head (SH), composed of two $3 \times 3$ convolutions followed by a $1 \times 1$ convolution. Acting as the final implicit decoder, the SH regresses the pixel-wise spectral intensity map $\boldsymbol{HSI}(\lambda) \in \mathbb{R}^{1 \times H \times W}$ at the continuous coordinate $\lambda$, as follows:

$$\boldsymbol{HSI}(\lambda) = \text{SH}(\text{Concat}(\boldsymbol{e}_\lambda, \boldsymbol{f})). \tag{16}$$

## 4. Experimental Results

**Datasets.** For continuous spectral reconstruction and spectral super-resolution, we utilize a non-redundant subset of 70 scenes from the ICVL dataset (Arad & Ben-Shahar, 2016), cropped to 163 bands across the 450-650 nm range. Specifically, this subset is randomly partitioned into 60 scenes for training and 10 scenes for testing. Regarding wavelength settings, we employ 143 sampled wavelengths for both training and evaluation in the continuous reconstruction task. For spectral super-resolution, the model is trained on these 143 bands and subsequently evaluated on the remaining 20 unseen wavelengths to verify its zero-shot synthesis capability.

For simulation experiments on the discrete reconstruction benchmark, consistent with prior works (Qin et al., 2025;

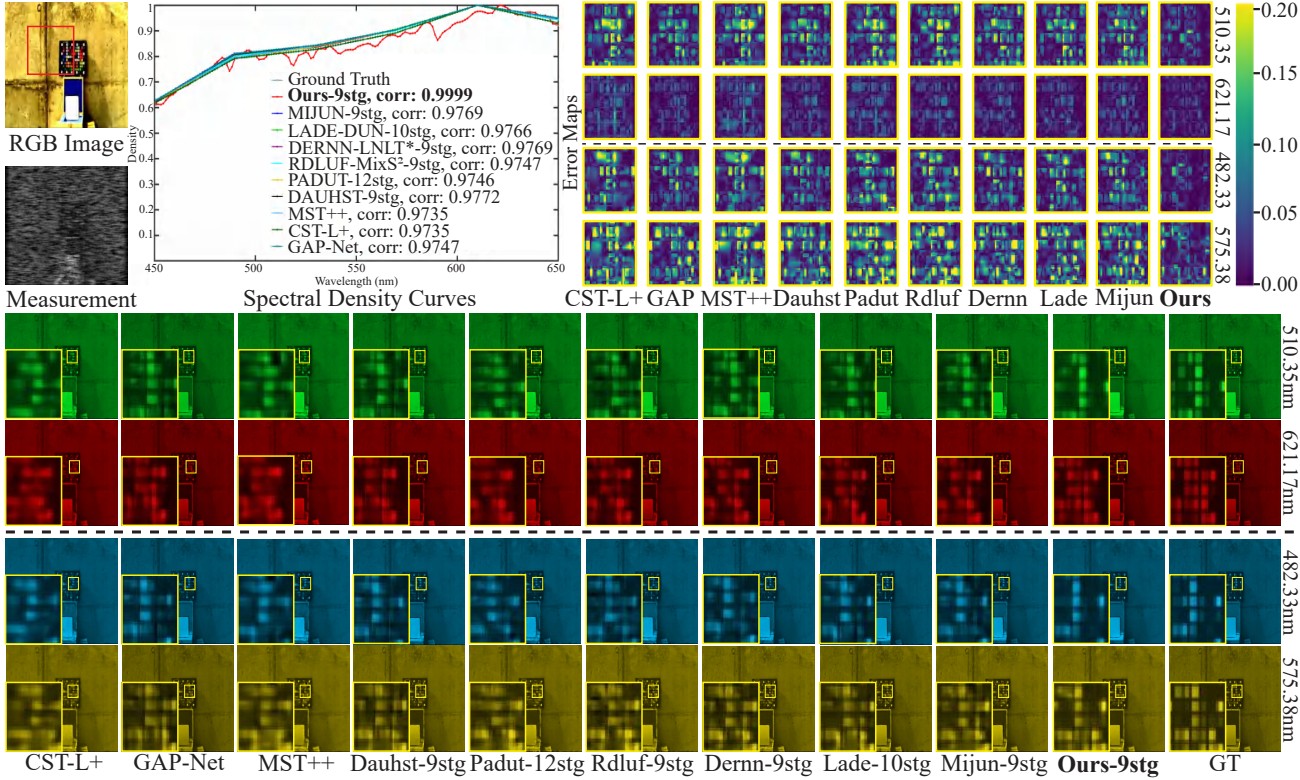

*Figure 5.* Qualitative comparison on ICVL scene IDS_COLORCHECK_1020-1215-1. The upper panel displays the reference RGB image, the 2D measurement, spectral density curves for the red-boxed region, and spatial error maps for the yellow-boxed regions. These maps visualize the absolute difference between each method and the ground truth (GT), where blue indicates lower errors and yellow signifies higher errors. The lower panel compares spectral slices: the first two rows show continuous spectral reconstruction at 510.35 nm and 621.17 nm, while the last two rows demonstrate spectral super-resolution at 482.33 nm and 575.38 nm.

Dong et al., 2024; Cai et al., 2022b), we adopt the CAVE (Huang & Shi, 2017) (32 scenes, 512×512) and KAIST (Choi et al., 2017) (30 scenes, 2704×3376) datasets, using CAVE for training and 10 scenes from KAIST for testing. For real experiments, the model is trained on the combined CAVE and KAIST datasets and tested on 5 real measurements (660×714×28) captured by a prototype CASSI system (Arce et al., 2013) covering the 450-650 nm range.

**Implementation Details.** During training, we adopt the Adam optimizer with a learning rate of $1 \times 10^{-3}$ and a cosine annealing scheduler. For continuous spectral reconstruction and super-resolution, a subset of 6 wavelengths is randomly sampled from the target spectral pool at each training iteration. The model is trained for 200 epochs on two NVIDIA A100 GPUs, each with 40 GB memory.

### 4.1. Continuous Spectral Reconstruction and Spectral Super-Resolution

We benchmark the proposed Phy-CoSF approach against nine state-of-the-art (SOTA) CASSI methods on ten representative scenes from the ICVL dataset. To ensure a comprehensive and fair evaluation, all competing methods are standardized by training on 6 wavelength channels. Since

*Table 2.* Quantitative comparison (average results) of spectral super-resolution on the ICVL dataset.

| Algorithms | SAM ↓ | PSNR ↑ | SSIM ↑ |
|---|---|---|---|
| MST++ (Cai et al., 2022b) | 2.36 | 34.52 | 0.885 |
| CST-L+ (Cai et al., 2022a) | 2.35 | 34.43 | 0.882 |
| GAP-Net (Meng et al., 2020a) | 2.31 | 36.06 | 0.914 |
| DAUHST-9stg (Cai et al., 2022c) | 2.25 | 35.81 | 0.911 |
| PADUT-12stg (Li et al., 2023) | 2.46 | 34.72 | 0.894 |
| RDLUF-MixS²-9stg (Dong et al., 2023) | 2.39 | 35.07 | 0.900 |
| DERNN-LNLT*-9stg (Dong et al., 2024) | 2.26 | 35.78 | 0.911 |
| Lade-DUN-10stg (Wu et al., 2024) | 2.08 | 35.85 | 0.914 |
| Mijun-9stg (Qin et al., 2025) | 2.29 | 35.31 | 0.902 |
| **Phy-CoSF-9stg** | **1.15** | **36.46** | **0.915** |

existing SOTA methods are predominantly designed for discrete spectral reconstruction, their 6-channel outputs are post-processed via interpolation to 143 and 20 channels during evaluation, enabling comparisons for continuous spectral reconstruction and spectral super-resolution, respectively. Notably, directly performing discrete reconstruction over 143 spectral channels would be computationally prohibitive in terms of memory footprint and processing time. Moreover, such an extreme 143:1 spectral compression ratio would incur significant information loss, severely degrading reconstruction accuracy.

In the comparative tables, we distinguish algorithm cate-

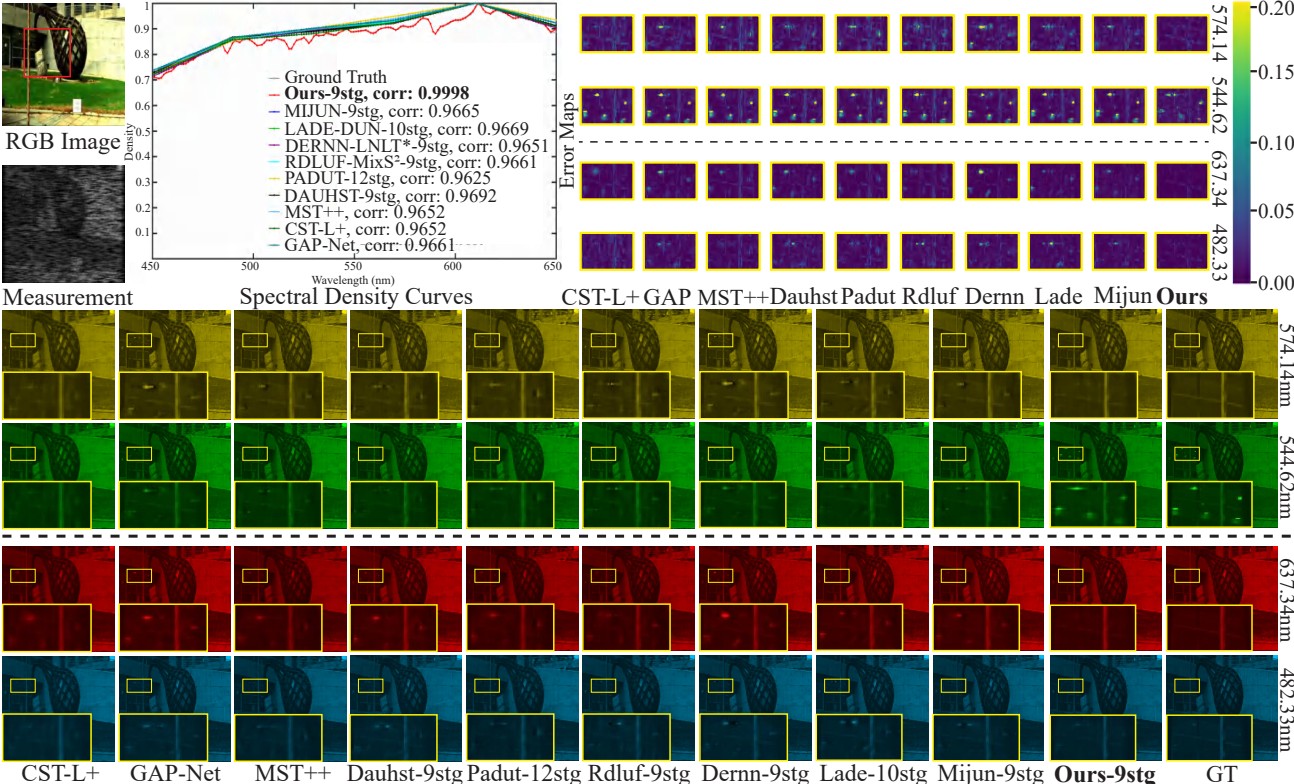

*Figure 6.* Qualitative comparison on ICVL scene bulb_0822-0909. The top two rows illustrate the results for continuous spectral reconstruction on training wavelengths. The bottom two rows demonstrate the results for spectral super-resolution on rendering wavelengths.

gories using a color-coding scheme: orange denotes end-to-end networks, while green represents deep unfolding networks. As shown in Table 1, Phy-CoSF achieves state-of-the-art continuous spectral reconstruction performance across all evaluated metrics and scenes, as evidenced by its significantly reduced SAM values, indicating exceptional spectral fidelity. Although our approach exhibits the highest FLOPs among 6-channel baselines, its computational complexity remains substantially lower than that of a discrete reconstruction model designed for 143-wavelength input. Table 2 further confirms that Phy-CoSF effectively maintains spectral consistency and spatial detail even on unseen wavelengths. As illustrated in Figure 5 and Figure 6, the visual results of Phy-CoSF are most consistent with the ground truth, characterized by improved spectral consistency and sharper textural structures, highlighting its superior capability in spatial detail restoration and precise spectral modeling. Additional experimental results are provided in the appendix.

### 4.2. Discrete Spectral Reconstruction

Table 3 summarizes the quantitative comparison on ten simulated scenes, where Phy-CoSF exhibits exceptional fidelity in restoring high-frequency details. It is worth noting that since the MGIR method (Li et al., 2025) is not publicly avail-

*Table 3.* Quantitative comparison of discrete spectral reconstruction on simulated data.

| Algorithms | PSNR ↑ | SSIM ↑ |
|---|---|---|
| MST++ (Cai et al., 2022b) | 35.72 | 0.955 |
| CST-L+ (Cai et al., 2022a) | 36.12 | 0.957 |
| MGIR (Li et al., 2025) | 35.69 | 0.951 |
| ADMM-Net (Ma et al., 2019) | 34.93 | 0.957 |
| GAP-Net (Meng et al., 2020a) | 32.89 | 0.919 |
| DGSMP (Huang et al., 2021) | 32.99 | 0.947 |
| DAUHST-9stg (Cai et al., 2022c) | 38.36 | 0.967 |
| PADUT-12stg (Li et al., 2023) | 38.89 | 0.974 |
| RDLUF-MixS$^2$-9stg (Dong et al., 2023) | 39.57 | 0.974 |
| **Phy-CoSF-9stg** | **39.80** | **0.978** |

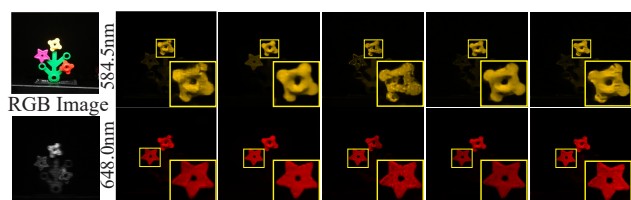

*Figure 7.* Qualitative comparison of real data. RGB image, measurement, and real discrete reconstruction results for scene 1 (2/28 channels).

able and is limited to simulation experiments, its results are cited directly from the original publication. Unlike tradi-

*Table 4.* Ablation study of different components for continuous spectral reconstruction on the ICVL dataset. We compare the proposed full model against variants with specific modules removed.

| Metrics | w/ Si | w/ Du | w/o SSH | w/o RFE | w/o SE | w/o FM | w/ SM | w/ All |
|---|---|---|---|---|---|---|---|---|
| SAM ↓ | 2.19 | 1.62 | 2.28 | 1.50 | 1.53 | 1.81 | 1.74 | **1.14** |
| PSNR ↑ | 34.63 | 36.01 | 35.42 | 36.32 | 35.55 | 35.94 | 36.03 | **36.45** |
| SSIM ↑ | 0.881 | 0.907 | 0.904 | 0.913 | 0.899 | 0.909 | 0.909 | **0.915** |

*Table 5.* Ablation study of different components for spectral super-resolution on the ICVL dataset. We compare the proposed full model against variants with specific modules removed.

| Metrics | w/ Si | w/ Du | w/o SSH | w/o RFE | w/o SE | w/o FM | w/ SM | w/ All |
|---|---|---|---|---|---|---|---|---|
| SAM ↓ | 2.04 | 1.58 | 2.20 | 1.47 | 1.49 | 1.77 | 1.70 | **1.15** |
| PSNR ↑ | 34.67 | 36.03 | 35.47 | 36.33 | 35.57 | 35.97 | 36.07 | **36.46** |
| SSIM ↑ | 0.881 | 0.907 | 0.904 | 0.912 | 0.900 | 0.909 | 0.909 | **0.915** |

tional discrete reconstruction methods that are restricted to fixed spectral bands, our approach is explicitly designed for continuous spectral reconstruction and spectral super-resolution, enabling it to more effectively capture the underlying physical structure of hyperspectral signals. This superiority extends to real measurements. Figure 7 presents the qualitative comparison exclusively under real scenarios. As observed, our approach yields sharper edges, superior spatial-spectral consistency, and significantly fewer artifacts than competing methods. These findings collectively validate the robustness of Phy-CoSF in handling complex spectral signals.

### 4.3. Ablation Studies

**Impact of Triple-Branch Cross-Domain Feature Mixer.** We evaluate three hierarchical variants: Single-branch (w/ Si), which utilizes only the native-resolution fine-grained feature $f_H$; Dual-branch (w/ Du), which integrates fine- and meso-scale features $(f_H, f_M)$; and Triple-branch (w/ All), which encompasses fine, meso, and coarse granularities $(f_H, f_M, f_L)$. As shown in Table 4 and Table 5, reconstruction performance consistently improves as the granularity hierarchy expands. Specifically, the introduction of meso- and coarse-scale branches provides broader context and semantic information that the high-resolution baseline lacks.

**Impact of Spectral Synthesis Head.** We evaluate four configurations: w/o SSH, which directly interpolates discrete reconstructions to target wavelengths; w/o RFE, which removes the random frequency encoding and uses raw coordinates; w/o SE, which omits the learnable spectral embedding; and the full SSH (w/ All). As illustrated in Table 4 and Table 5, w/o SSH fails to capture fine-grained spectral variations, and removing RFE leads to a noticeable performance drop, confirming its role in overcoming spectral bias toward low frequencies. Furthermore, the SE module is crucial for high-fidelity continuous spectral synthesis by adapting fixed Fourier features to specific spectral signatures.

**Impact of Fourier Mamba.** We evaluate three variants: w/o Fourier Mamba (w/o FM), which removes the frequency-domain sequence modeling branch; Spatial Mamba (w/ SM), which applies Mamba directly in the spatial domain; and the proposed full model (w/ All). As illustrated in Table 4 and Table 5, the FM consistently outperforms the SM and the baseline. While SM captures long-range relations via spatial scanning, the proposed FM leverages the 2D FFT to transform spatial information into compact global frequency coefficients, enabling more effective modeling of global structural priors and inter-frequency relationships.

## 5. Discussion, Limitations and Future Work

Our proposed Phy-CoSF successfully establishes a new paradigm for snapshot compressive imaging, overcoming the fixed-band limitations of previous discrete models to achieve continuous spectral reconstruction and arbitrary spectral super-resolution. However, our method is not without limitations. Although Phy-CoSF is substantially more efficient than 143-wavelength discrete models, it still exhibits the highest FLOPs among all 6-channel baselines. This high computational demand may constrain its real-time deployment on resource-limited edge devices. We believe that introducing multi-resolution hash grids to optimize the spectral synthesis head can significantly reduce computational overhead, thereby effectively enhancing the inference efficiency and practicality of the model. Furthermore, we believe this continuous physics-guided unfolding framework opens exciting new avenues for other high-dimensional inverse problems and holds the promise of being readily extended to tasks such as video-spectral snapshot compressive imaging.

## 6. Conclusion

In this paper, we proposed Phy-CoSF, a novel framework that synergized physics-guided DUNs with continuous INRs to overcome the discrete limitations of traditional architectures. We established a two-phase paradigm enabling discrete-band training while supporting high-fidelity rendering at arbitrary continuous resolutions. This capability was achieved by designing a continuous spectral fields that replaced the discrete prior. It featured a triple-branch cross-domain encoder that fused spatial, channel, and mamba-based frequency features to distill robust, wavelength-agnostic representations. These were decoded by a spectral synthesis head to generate intensities at continuous coordinates. Extensive experiments demonstrated the superior reconstruction fidelity of Phy-CoSF, establishing a new standard for physics-guided continuous spectral reconstruction.

## Acknowledgements and Disclosure of Funding

This research is supported in part by National Natural Science Foundation of China under Grant 62471199, 62020106011, and 62271414, in part by National Foreign Experts Program under Grant S20250222, in part by National Natural Science Fund for Excellent Young Scientists Fund Program (Overseas), in part by National Key R&D Program Project under Grant 2023YFC3806003, in part by Scientific Research Project of the Education Department of Jilin Province under Grant JJKH20261279KJ, and in part by Science and Technology Development Program of Jilin Province under Grant 20260205063GH.

## Impact Statement

Phy-CoSF establishes a new paradigm for snapshot compressive imaging by synergizing physics-guided deep unfolding with implicit neural representations to achieve continuous spectral reconstruction. Unlike traditional discrete-band methods, our train-render framework enables high-fidelity arbitrary spectral super-resolution in a zero-shot manner, significantly enhancing data reliability for remote sensing and medical diagnostics. By integrating a Mamba-based module for efficient global context modeling, this work ensures physical consistency and superior reconstruction accuracy, providing a robust and scalable AI solution for high-dimensional imaging challenges.

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

## A. Additional Experimental Details

*Table 6.* The specific scene names from the ICVL dataset utilized for the training and rendering phases.

| Train-Render Phases | Scene Names |
|---|---|
| Training Scenes | 4cam_0411-1648, bgu_0403-1439, bgu_0403-1444, bgu_0403-1525, BGU_0522-1113-1, BGU_0522-1136, BGU_0522-1201, BGU_0522-1203, BGU_0522-1211, BGU_0522-1216, bguCAMP_0514-1712, bguCAMP_0514-1718, bguCAMP_0514-1723, bguCAMP_0514-1724, eve_0331-1549, eve_0331-1601, eve_0331-1602, eve_0331-1606, eve_0331-1618, eve_0331-1633, eve_0331-1656, Flower_0325-1336, gavyam_0823-0930, gavyam_0823-0933, gavyam_0823-0944, grf_0328-0949, Labtest_0910-1502, Labtest_0910-1504, Labtest_0910-1509, Labtest_0910-1510, Labtest_0910-1513, lehavim_0910-1600, lehavim_0910-1602, lehavim_0910-1607, lehavim_0910-1610, Lehavim_0910-1622, Lehavim_0910-1627, Lehavim_0910-1629, Lehavim_0910-1630, Lehavim_0910-1633, Lehavim_0910-1708, Lehavim_0910-1716, lst_0408-0950, lst_0408-1004, lst_0408-1012, Master5000K_2900K, Master20150112_f2_colorchecker, maz_0326-1048, Maz0326-1038, nachal_0823-1038, nachal_0823-1040, nachal_0823-1047, nachal_0823-1110, nachal_0823-1117, nachal_0823-1118, nachal_0823-1127, nachal_0823-1149, nachal_0823-1152, nachal_0823-1223, negev_0823-1003 |
| Rendering Scenes | BGU_0403-1419-1, BGU_0522-1217, bulb_0822-0909, eve_0331-1646, hill_0325-1242, IDS_COLORCHECK_1020-1215-1, Labtest_0910-1511, Lehavim_0910-1636, Lehavim_0910-1718, nachal_0823-1144 |

### A.1. Implementation Details

For inputs containing six spectral wavelengths, the channel dimension $C$ of the wavelength-agnostic latent representation $f$ is 72. The spatial resolution is standardized to $H = W = 256$ in both the training and rendering phases, which is achieved via random patch cropping during training.

Regarding architectural configurations, the downsampling module in the CoSF is implemented using a convolutional layer with a kernel size of $4 \times 4$ and a stride of 2. For the random frequency encoder (RFE) module, the mapping dimension $m$ is set to 32, and the Gaussian standard deviation $\sigma$ is fixed to 1. In addition, the latent embedding dimension $D$ of the spectral embedding (SE) module is set to 64.

### A.2. Dataset Details

To rigorously evaluate the continuous spectral reconstruction and spectral super-resolution capabilities of the proposed Phy-CoSF, we conduct extensive experiments on the widely used the ICVL dataset (Arad & Ben-Shahar, 2016). The detailed partitioning strategy for training and rendering scenes is summarized in Table 6. For clarity, the scene identifiers (e.g., Scene1, Scene2) reported in Table 1 correspond sequentially to the rendering scenes listed in Table 6.

A key characteristic of our experimental design is the explicit decoupling of training and rendering wavelengths. Specifically, we consider a total of 163 spectral bands covering the wavelength range from 450 nm to 650 nm. Among them, 20 bands are reserved as unseen wavelengths to evaluate spectral super-resolution performance, namely: 452.02 nm, 461.71 nm, 471.40 nm, 482.33 nm, 492.06 nm, 503.03 nm, 512.79 nm, 522.56 nm, 533.58 nm, 543.39 nm, 554.44 nm, 564.29 nm, 575.38 nm, 585.25 nm, 595.14 nm, 606.28 nm, 616.20 nm, 627.38 nm, 637.34 nm, and 648.55 nm. These wavelengths are randomly distributed across the spectral range to avoid selection bias. The remaining 143 bands are used for training.

## B. Additional Experimental Results

### B.1. Continuous Spectral Reconstruction and Spectral Super-Resolution

For continuous spectral reconstruction, as illustrated in Figure 5 and Figure 6, our approach consistently achieves superior performance, producing spectral intensity curves that align most closely with the ground truth (GT). In contrast to conventional pipelines that perform discrete reconstruction followed by interpolation, which are prone to error accumulation and unreliable spectral estimation due to their decoupling from the physical measurement $y$, the proposed Phy-CoSF

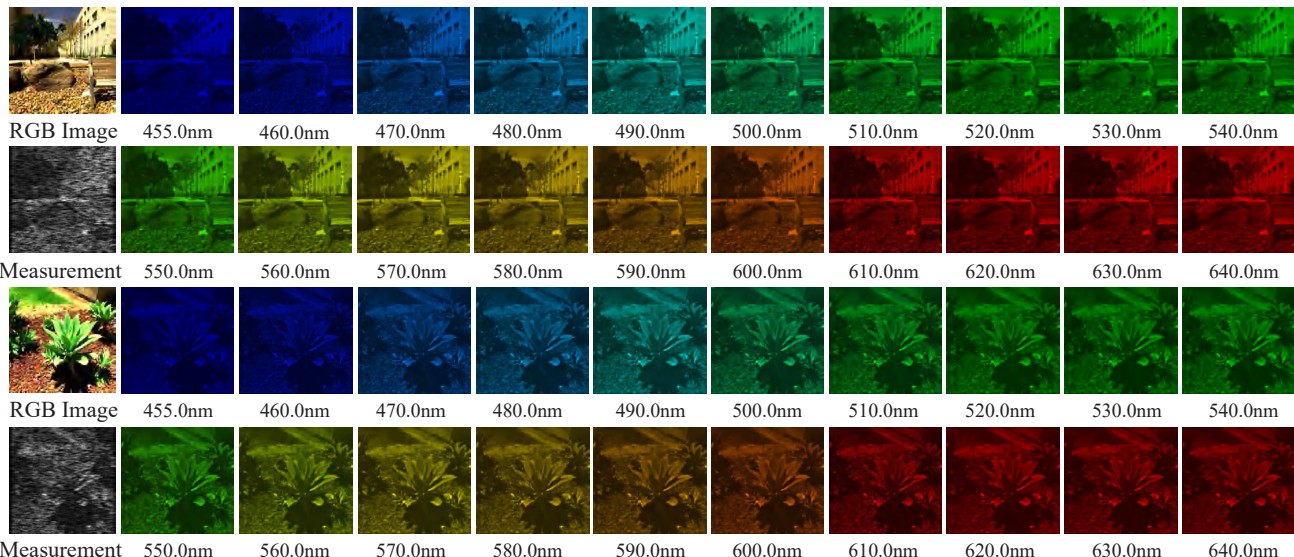

*Figure 8.* Additional spectral super-resolution results on ICVL scenes BGU_0403-1419-1 and BGU_0522-1217. We specifically render images at novel spectral coordinates distinct from the standard 163 wavelengths, demonstrating the capability of the model to generalize to unseen spectral bands.

*Table 7.* Quantitative comparison (average results) on a subset of 28 randomly sampled wavelengths from the ICVL dataset.

| Continuous Spectral Reconstruction | SAM ↓ | PSNR ↑ | SSIM ↑ |
|---|---|---|---|
| Mijun-5stg (Qin et al., 2025) | 1.64 | 37.31 | 0.927 |
| **Phy-CoSF-5stg** | **1.26** | **37.56** | **0.932** |
| Spectral Super-Resolution | SAM ↓ | PSNR ↑ | SSIM ↑ |
| Mijun-5stg (Qin et al., 2025) | 1.65 | 37.31 | 0.928 |
| **Phy-CoSF-5stg** | **1.24** | **37.58** | **0.932** |

integrates continuous spectral modeling directly into the physics-guided reconstruction loop. As a result, it exhibits high fidelity in preserving spectral distributions while accurately recovering fine-grained spatial details that are often lost under coarse discrete approximations.

Furthermore, we report additional results on spectral super-resolution in Figure 8, where the model synthesizes images at novel wavelength coordinates beyond the fixed 163 training and standard rendering bands. These results empirically demonstrate that Phy-CoSF is capable of extracting intrinsic sub-band spectral information, rather than relying on statistical interpolation. By conditioning the continuous spectral query on the compressed measurement, our method ensures robust spectral consistency and precise structural recovery even at previously unseen wavelengths, while strictly adhering to data fidelity constraints that post-processing-based approaches fail to enforce. For a more comprehensive evaluation, we provide additional results on both continuous spectral reconstruction and spectral super-resolution across all 10 test scenes from the ICVL dataset in the supplementary material.

Moreover, existing discrete baselines are inherently restricted by their fixed output architectures, rendering them incompatible with the random spectral sampling strategy. Attempting to train these rigid models with stochastic wavelengths disrupts the learned spectral correlations, leading to convergence failure and severe performance degradation. In contrast, our coordinate-based approach flexibly adapts to arbitrary spectral queries. Fundamentally, interpolating from either 6 or 28 discrete bands lacks true physical meaning. We deliberately chose the 6-band setting to more explicitly expose the flaws of existing methods and to strictly match our per-iteration sampling size, thereby ensuring perfectly identical hardware constraints for a fair comparison. Furthermore, to provide a more comprehensive evaluation, we trained the baseline model on 28 bands and interpolated its discrete outputs to 143 and 20 bands. As shown in Table 7, our continuous method, Phy-CoSF, still consistently outperforms the baseline method.

To model complex spectral dependencies, recent state-of-the-art methods commonly rely on dense spectral attention or correlation modules, which inherently incur a quadratic computational complexity of $\mathcal{O}(N^2)$. Based on their reported

*Table 8.* Complexity comparison in terms of parameters and FLOPs at 143 spectral bands.

| Methods | Params (M) | FLOPs (G) |
|---|---|---|
| DAUHST-9stg (Cai et al., 2022c) | 25.65 | 1630.0 |
| PADUT-12stg (Li et al., 2023) | 31.99 | 2249.0 |
| RDLUF-MixS²-9stg (Dong et al., 2023) | 11.19 | 1980.0 |
| Mijun-9stg (Qin et al., 2025) | 3.28 | 1593.0 |
| Lade-DUN-10stg (Wu et al., 2024) | 10.88 | 2036.0 |
| **Phy-CoSF-9stg** | **0.27** | **801.38** |

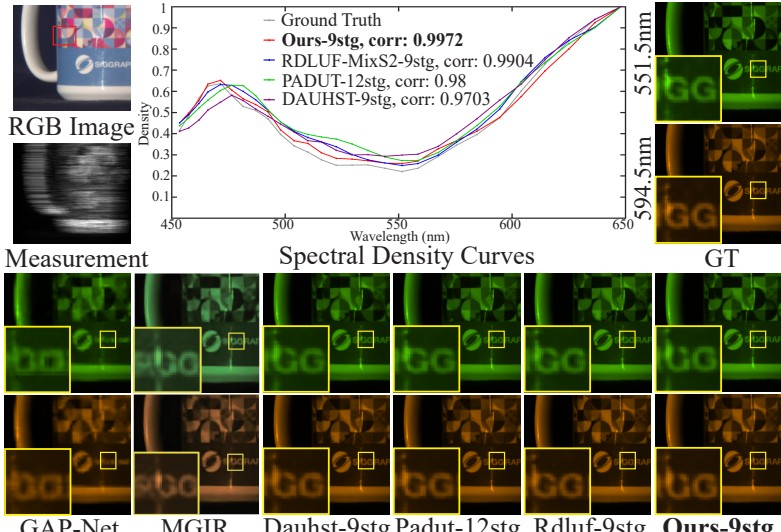

*Figure 9.* Qualitative comparison of simulated data. RGB image, measurement, and spectral density curves (red box) for scene 5, followed by simulated discrete reconstruction results (2/28 channels).

performance at 6 and 28 spectral bands, we conduct a regression-based analysis to extrapolate their computational costs at 143 bands, as summarized in Table 8. The results reveal a pronounced scalability bottleneck: computational costs increase dramatically, with LADE-DUN (Wu et al., 2024) and PADUT (Li et al., 2023) projected to exceed 2,000 G FLOPs, rendering them impractical under realistic memory constraints. Moreover, the extreme compression ratio of 143:1 further aggravates the reconstruction challenge, as recovering dense spectral information from a single compressed snapshot becomes severely ill-posed. In contrast, Phy-CoSF exhibits substantially improved efficiency, requiring only 0.27 M parameters and 801.38 G FLOPs to process the same 143-band input. These results indicate that conventional discrete architectures are feasible only at limited spectral resolutions, whereas the proposed Phy-CoSF provides a scalable and practical solution for high-spectral-resolution reconstruction.

### B.2. Discrete Spectral Reconstruction

Qualitative comparisons are presented in Figure 9. As observed, Phy-CoSF exhibits clear advantages in recovering fine-grained spatial details and preserving spectral fidelity. We further emphasize the architectural benefits of the proposed physics-guided deep unfolding framework over purely end-to-end (E2E) approaches such as MGIR (Li et al., 2025). While E2E models typically rely on black-box mappings and may suffer from limited physical consistency, the integration of deep unfolding with implicit neural representations in Phy-CoSF enforces data fidelity and physical interpretability throughout the reconstruction process. This structured prior allows Phy-CoSF to substantially outperform MGIR, particularly in challenging scenarios involving precise spectral interpolation and generalization to previously unseen wavelengths.

### B.3. Ablation Studies

To provide a holistic evaluation of our proposed components, we present a detailed visual comparison for both continuous spectral reconstruction and spectral super-resolution in Figure 10. As shown in the top two rows of Figure 10, we first

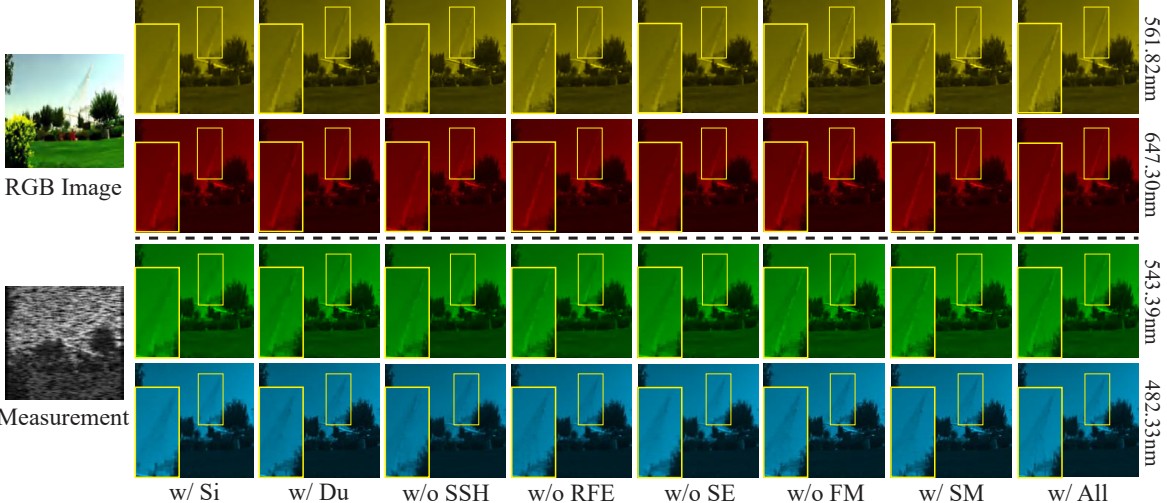

*Figure 10.* Additional ablation studies on ICVL scene nachal_0823-1144. The top two rows illustrate the results for continuous spectral reconstruction on training wavelengths. The bottom two rows demonstrate the results for spectral super-resolution on rendering wavelengths.

examine the reconstruction quality on standard training wavelengths. The ablated variants exhibit noticeable degradation: the single-branch feature mixer (w/ Si) often yields blurred spatial textures, while the removal of the Fourier Mamba (w/o FM) results in structural inconsistencies. In contrast, the full Phy-CoSF model (w/ All) effectively suppresses artifacts and recovers sharp edges, demonstrating superior fidelity in the standard reconstruction setting.

To rigorously validate the zero-shot generalization capability, the bottom two rows of Figure 10 illustrate the performance on synthesizing novel wavelengths not seen during training, where the contribution of each module becomes even more pronounced. Regarding the cross-domain feature mixer, the single-branch baseline struggles to infer spatial semantics at unseen coordinates, whereas the introduction of meso- and coarse-scale branches (w/ All) significantly enhances structural coherence by leveraging multi-scale context. Similarly, for the spectral synthesis head (SSH), the removal of random frequency encoding (w/o RFE) leads to severe spectral over-smoothing, failing to recover high-frequency chromatic variations; however, the full SSH preserves high-fidelity intensity distributions even for unobserved wavelengths. Finally, in terms of Fourier Mamba, while the Spatial Mamba (w/ SM) effectively captures local dependencies, it fails to maintain global consistency. In contrast, the Fourier Mamba leverages global frequency interactions to ensure structural integrity across the continuous spectrum. Correspondingly, as shown in Table 4 and Table 5, the quantitative metrics are highly consistent with our visual findings. Specifically, the full Phy-CoSF model achieves the best SAM, PSNR, and SSIM scores across all unseen wavelengths, confirming that the synergy of these components is essential for achieving robust spectral generalization.

