# OpenReview forum: "Phy-CoSF: Physics-Guided Continuous Spectral Fields Reconstruction and Spectral Super-Resolution for Snapshot Compressive Imaging"
_ICML.cc/2026/Conference — ICML 2026 regular_

### Official Review · Reviewer_aZHz · 2026-02-25

**Soundness:** 3
**Presentation:** 4
**Significance:** 3
**Originality:** 3
**Overall Recommendation:** 5
**Confidence:** 3

**Summary:**

This paper proposes a unrolling network for continuous spectral reconstruction. The main contributions are as follows:
1. The authors introduce a train-render two-phase paradigm, where the model is trained on randomly sampled discrete wavelengths but can render spectral outputs at arbitrart continuous wavelengths during validating.
2. The Fourier Mamba module is incorporated into the continuous spectral modeling to capture global frequency-domain information.

**Compliance With Llm Reviewing Policy:**

Affirmed.

**Final Justification:**

The experimental results are comprehensive and the rebuttal is convincing, although I still think the core DUN framework has already been utilized in other application fields. On this point, I share the same view as Reviewer THnz. Overall, in the final justification, I increase the overall score from 4 to 5 and increase the presentation score from 3 to 4.

**Key Questions For Authors:**

1. Although this paper utilizes the unrolling framework to design network architecture, which is mathematically interpreted, it lacks theoretical analysis, especially the convergence analysis. Moreover, the unrolling framework is not very novel. Before this work, many works explores the unrolling framework in the field of computational imaging.

2. The competing methods are trained on 6 discrete wavelengths and their outputs are interpolated to 143 (or 20) bands for continuous evaluation. Since these methods were not originally designed for continuous modeling, how do the authors justify that this protocol provides a fair comparison?

3. The train-render paradigm assumes training on randomly sampled discrete wavelengths within a fixed spectral range. How robust is the learned continuous representation when the spectral distribution changes (e.g., different sampling density, different spectral response functions, or different CASSI hardware settings)?

**Limitations:**

yes

**Strengths And Weaknesses:**

Strength:
1. This paper addresses an important limitation of exisfing CASSI reconstruction methods, namely their restriction to fixed discrete spectral bands. Moving toward continuous spectral reconstruction and arbitrary wavelength querying is conceptually meaningful and better aligned with the physical continuity of hyperspectral signals.

2. The proposed decoupling between discrete-band training and continuous rendering is well motivated. It enables continuous spectral synthesis without requiring dense supervision over all wavelengths during training.

Weakness:
1. While the physics motivation is meaningful and the unrolling network is constructed based on the optimization algorithm, the paper lacks theoretical analysis, such as the covergence analysis.
2. Competing methods are originally designed for discrete reconstruction and are adapted via interpolation for continuous evaluation. This may not fully reflect their intended capabilities, potentially affecting the fairness of comparison.
3. Many core blocks (deep unfolding, implicit representations, Fourier features, Mamba modules) in the proposed framework are existing techniques.

---

> ### Author Rebuttal · Authors · 2026-03-31
>
> Thank you for your detailed review and for highlighting the strengths of our work. We have carefully considered your comments and offer our responses below.
>
> **Q1**: Theoretical and convergence analysis
>
> **Response**: We thank the reviewer for this rigorous suggestion. Theoretically, the underlying A-HQS optimization algorithm possesses strict mathematical convergence guarantees. While embedding a deep continuous prior (CoSF) introduces non-linearity, the explicit data-fidelity constraint applied at each unfolding stage tightly bounds the solution space, ensuring strong convergence.
>
> To validate this, we have provided an convergence curve in the Fig. 6 (link: https://anonymous.4open.science/r/Phy-CoSF), which clearly demonstrates the stable and rapid convergence of our model during training. We agree that this is a valuable addition; if accepted, we will explicitly include this convergence analysis and a theoretical discussion of the A-HQS unrolling mechanism in the revised manuscript.
>
> **Q2**: Fairness of evaluating discrete baselines on continuous tasks
>
> **Response**: We appreciate the reviewer’s perspective and explicitly acknowledge that existing baselines were designed for discrete reconstruction tasks. However, this is precisely the fundamental limitation our work aims to address.
>
> 1. Interpolation: Because discrete models inherently lack the capability to query arbitrary or dense wavelengths, mathematical interpolation is the only feasible way to adapt them for continuous evaluation. Directly scaling their architectures to natively output 143 bands causes intractable Out-of-Memory failures and computational explosion, as analyzed in *Table* 5 (manuscript).
>
> 2. Fairness on Tasks: To ensure absolute fairness, we did rigorously evaluate all baselines on their originally intended discrete settings (the standard 28-band CAVE/KAIST benchmarks). As shown in *Figure* 6 and *Table* 6 (manuscript), our Phy-CoSF still achieves state-of-the-art results under their native protocol.
>
> In short, the continuous evaluation protocol is not designed to unfairly penalize the baselines. Rather, it objectively demonstrates a crucial new capability: when arbitrary spectral super-resolution is required, traditional discrete paradigms are fundamentally inadequate, whereas our continuous physics-guided framework natively excels.
>
> **Q3**: Novelty of techniques
>
> **Response**: We acknowledge that concepts like DUN, INRs, Fourier and Mamba exist in computational imaging. However, Phy-CoSF is not a simple plug-and-play stacking, but a targeted structural reinvention for the highly ill-posed CASSI system:
>
> 1. Continuous Prior in DUN: While standard unrolling relies on discrete priors, our fundamental novelty is embedding an CoSF module as a dynamic continuous prior directly inside a physics-guided loop. This uniquely guarantees data-fidelity for arbitrary wavelength queries.
>
> 2. Handling Physical Dispersion: Standard Fourier or Mamba modules cannot natively resolve CASSI's severe spatial-spectral entanglement. We explicitly designed the CDFE to cascade spatial, frequency, and channel representations to overcome this specific physical dispersion.
>
> Ultimately, the novelty of Phy-CoSF lies in pioneering a fundamentally new continuous optimization paradigm for CASSI, intrinsically coupling physics-guided unfolding with implicit spectral representations to achieve arbitrary-resolution reconstruction for the first time.
>
> **Q4**: Robustness to varying sampling densities and CASSI hardware
>
> **Response**: We thank the reviewer for this insightful suggestion. To explicitly evaluate the robustness of our learned continuous representation, we conducted two additional sets of experiments:
>
> 1. Varying Sampling Density: We increased the training sampling density from 6 to 28 bands. As shown in the linked Tab.4 and Fig.4, our method maintains highly stable and superior performance in both continuous reconstruction and super-resolution, proving it adapts robustly to different spectral sampling distributions.
>
> 2. Varying CASSI Hardware: By evaluating our model using completely unseen physical masks (injected with random noise) during testing, we explicitly simulate hardware-level physical variations, such as different spectral response functions. This proves that Phy-CoSF learns a genuine, physically meaningful continuous spectral prior, rather than merely overfitting to the photoelectric conversion curve of a specific camera. As demonstrated in the linked Fig. 7 and Tab. 5 (link: https://anonymous.4open.science/r/Phy-CoSF), if new physical parameters are fed into our model, it can still robustly adapt and maintain high-quality spectral reconstruction.

---

> > ### Author Rebuttal · Reviewer_aZHz · 2026-04-01
> >
> > 1. While the authors argue that Phy-CoSF is a structural reinvention, the current description still appears to be a composition of existing components (deep unfolding, INR, Fourier features, and Mamba-like modules). It remains unclear what fundamentally new modeling capability is introduced beyond a carefully engineered integration.
> > 2. Unrolling-based algorithms relies heavily on the correctness of the assumed optimization model, making it sensitive to model mismatch and limiting its generalization ability in real-world scenarios. Moreover, the optimization algorithm underlying unrolling methods is typically derived under specific constraints, yet after being unfolded into a neural network, these constraints are not necessarily preserved, which may lead to a mismatch between the theoretical formulation and the learned model. How do you think of the potentail development of the unrolling methods?
> > 3. From the persepctive of experiments, the authors have addressed all my concerns.

---

> > > ### Author Response · Authors · 2026-04-02
> > >
> > > Thank you very much for your follow-up questions and for the detailed review of our paper.
> > >
> > > **Q1**: Fundamentally new modeling capability
> > >
> > > **Response**: We sincerely apologize if our description did not clearly highlight our core contributions. We respectfully clarify that Phy-CoSF introduces fundamentally new modeling capabilities from both theoretical and architectural perspectives:
> > >
> > > 1. Theoretical Level
> > >
> > > For the first time, we enable iterative physical optimization directly on a continuous spectral space for CASSI. This establishes a pioneering paradigm in this field: training on discrete bands while rendering at arbitrary continuous wavelengths.
> > >
> > > · The Fundamental Shift: Traditional deep unfolding can only optimize discrete tensors and cannot use a discrete physical mask to directly constrain a continuous function. To overcome this, we fundamentally reformulated the proximal mapping stage. This unique formulation grants the network the capability to strictly and iteratively project continuous, coordinate-based representations back into the discrete physical measurement space at every unrolled stage.
> > >
> > > 2. Architectural Level
> > >
> > > While built upon a DUN framework, our internal modules are fundamentally novel, specifically within the proposed CoSF module:
> > >
> > > · Triple-Branch Cross-Domain Feature Mixer: Unlike the standard U-Net backbones used in prior HSI works, our Mixer fundamentally alters feature processing. By interpolating features from different domains into a unified dimension, it enables deep, comprehensive cross-domain fusion to explicitly resolve the severe spatial-spectral entanglement inherent to CASSI.
> > >
> > > · Spectral-Spatial Head (SSH): While conceptually inspired by INRs, the specific internal architecture of the SSH (including the SH and SE components) and its integration with the Mixer's features are entirely novel. Notably, the SH component completely replaces the traditional, simplistic MLP output heads found in standard INRs, providing a specialized mapping tailored specifically for high-fidelity continuous hyperspectral generation.
> > >
> > > Furthermore, the effectiveness of these proposed modules is explicitly validated by the ablation studies presented in *Table* 3, *Table* 7, and *Figure* 9 (manuscript).
> > >
> > > **Q2**: Development of the unrolling methods
> > >
> > > **Response**: We highly appreciate this insightful comment identifying two fundamental bottlenecks in current DUNs: real-world model mismatch and the loss of theoretical guarantees. We believe the field must evolve from rigidly unfolded to adaptively constrained frameworks, specifically in three directions:
> > >
> > > 1. Adaptive Unrolling: To address model mismatch (like CASSI hardware calibration shifts), future DUNs should dynamically estimate physical degradation parameters (like mask deviations) within the optimization loop. This enables self-calibration during inference, significantly enhancing real-world generalization.
> > >
> > > 2. Strictly Constrained Neural Operators: Connecting theoretical formulations with learned models requires mathematically constraining the opaque nature of deep priors. Imposing specific bounds, including Lipschitz continuity or explicit projection layers, allows neural architectures to maintain the exact convergence guarantees of the original optimization algorithms.
> > >
> > > 3. Hybridizing Physics with Expressive Priors: Since static physical models cannot capture all real-world complexities, DUNs require highly expressive priors to absorb unmodeled dynamics. This aligns precisely with our work: embedding an INR as a dynamic continuous prior bridges rigid deterministic equations with complex real-world continuous distributions.
> > >
> > > **Q3**: Experiments
> > >
> > > **Response**: We sincerely thank the reviewer for this highly positive feedback and for recognizing the rigor of our experimental efforts. Your constructive and insightful comments throughout the review process have been invaluable in helping us comprehensively validate our approach and significantly elevate the quality of our manuscript.
> > >
> > > **Additional Clarification on the DUN Framework**: Regarding your observation in the final justification, we completely agree that DUN is a well-established framework and we do not claim it as our core novelty. In fact, state-of-the-art methods in our field predominantly rely on this architecture. We utilize it as an indispensable physical constraint: unlike standard black-box networks that operate blindly and generate mathematical hallucinations, DUN explicitly brings the physical camera mask into the optimization loop. This mechanism forces every generated continuous spectrum to strictly obey real-world physical measurement constraints. Our true novelty lies in lifting this established discrete optimization into a continuous function space.

---

### Official Review · Reviewer_A6Jp · 2026-03-10

**Soundness:** 2
**Presentation:** 3
**Significance:** 2
**Originality:** 3
**Overall Recommendation:** 4
**Confidence:** 3

**Summary:**

This paper proposes a neural implicit representation for snapshot compressive hyperspectral imaging for continuous instead of discrete spectral reconstruction. The model is trained on discrete spectral bands and can render arbitrary continuous bands, enabling spectral super-resolution. The paper includes numerous comparisons against existing discrete-sampling methods, including both end-to-end and deep unfolding networks. The proposed architecture itself is based on deep unfolding with a degradation aware module and continuous spectral fields module. The method is evaluated on ten scenes from the ICVL dataset, showing improvements across multiple scenes.

**Compliance With Llm Reviewing Policy:**

Affirmed.

**Final Justification:**

Based on the rebuttal, I have increased my score. The rebuttal addressed my concerns about the evaluation and presented additional results comparing performance against methods trained to recover 28 bands, as well as detailed spectral profiles of the held-out spectral bands.

I have updated my score to a weak accept. While the paper has clear merits, its focus is confined to a narrow application (compressive hyperspectral imaging with CASSI), which limits its broader impact within the machine learning community.

**Key Questions For Authors:**

Could the authors please clarify on the evaluation approach in the paper (see strengths and weaknesses comments). In particular, I would be willing to change my evaluation with a clarification and more information about:
1) How to fairly compare against prior methods when the proposed method is trained on 143 bands and the prior methods are trained on 6 bands and then interpolated to 143. The prior methods originally were trained on 28 bands, not 6.
2) Spectral super-resolution evaluation. Please show spectral profiles that highlight the unseen bands and compare them with simple interpolation methods.
3) Address the PSNR discrepancy between prior works.

**Limitations:**

The limitations of spectral super-resolution are not addressed. Spiky, non-smooth spectra may be missed. No analysis is provided of failure cases for spectral super-resolution, or the potential consequences of relying on super-resolution for critical tasks.

**Strengths And Weaknesses:**

Soundness:
The technical section seems sound, but I did not individually check all of the equations.
The evaluation section does not seem sound to me and has several key issues.
1) Unless I am mistaken, the proposed method is trained on 143 different wavelengths and the prior method comparisons are trained on 6 wavelength channels and then interpolated to 143 and 20 channel outputs. This is not a fair comparison. Most of the other methods are originally trained on 28 bands (not 6!). E.g. both Qin et. al and Wu et al. use 28 bands. This choice of using 6 bands instead of the original 28 bands for the comparisons is not justified.
2) Prior work (e.g. Qin et. al. and Wu et. al) reports PSNRs around 40dB, but achieves significantly lower PSNR in the paper. This discrepancy should be addressed.
3) The goal of spectral super-resolution is not thoroughly evaluated. I would expect to see a result figure with spectral profiles that highlight which bands were in the training set and which were held out. The method’s prediction for these queried (and unseen) bands should be compared against a simple interpolation scheme. In the existing results figures, it is unclear which of the wavelengths were used for training vs. not. In addition, I would expect to see quantitative metrics for just the 20 unseen wavelengths to separate the overall reconstruction quality from the rendering-phase super-resolution performance.

Presentation:
The paper is clearly written and structured.
The paper does not mention some relevant prior work in implicit neural representations for compressive hyperspectral imaging. The paper claims that “To the best of our knowledge, this is the first work to jointly address continuous spectral reconstruction and super-resolution via a unified framework for CASSI”, but there are a few other papers that accomplish similar things with INRs. For example:
- Zhang, Kaiwei, et al. "Implicit neural representation learning for hyperspectral image super-resolution." IEEE Transactions on Geoscience and Remote Sensing 61 (2022): 1-12.
- Chen, Huan, et al. "Spectral-wise implicit neural representation for hyperspectral image reconstruction." IEEE Transactions on Circuits and Systems for Video Technology 34.5 (2023): 3714-3727.

Significance:
The paper addresses a problem that is narrow in scope and only applicable to the compressive hyperspectral imaging community. This work is domain-specific and quite specialized.

Originality:
The paper proposes a novel unrolling method with an implicit continuous representation. This seems original in the context of compressive hyperspectral imaging.

---

> ### Author Rebuttal · Authors · 2026-03-31
>
> Thank you for your thoughtful comments, constructive feedback, and for acknowledging the merits of our work. We will address each of them in turn below.
>
> **Q1**: Fairness of comparison and 6 vs 28 bands
>
> **Response**: We apologize for the confusion and clarify that existing methods are inherently restricted to fixed-wavelength reconstruction.
>
> 1. 143 bands: Phy-CoSF randomly samples 6 bands per iteration from a 143-band pool. Existing models require fixed-channel outputs: applying our random sampling causes convergence failure, while scaling them to 143 bands simultaneously triggers Out-of-Memory errors (*Table* 5 in the manuscript). The 6-band setup was strictly constrained by the baselines' architectural bottlenecks.
>
> 2. Why 6 bands: Fundamentally, interpolating from either 6 or 28 discrete bands lacks true physical meaning. We deliberately chose 6 bands to more explicitly expose the flaws of existing methods, and to strictly match our per-iteration sampling size, ensuring perfectly identical hardware constraints for a fair comparison.
>
> 3. 28-Band Evaluation: The 28-band setting of existing methods specifically targets discrete tasks. Accordingly, for the standard discrete benchmark in our manuscript, we also evaluated all methods on the original 28 bands, as shown in *Figure* 6 and *Table* 6.
>
> 4. New 28-Band Continuous Experiment: To completely resolve your concern, we trained the baselines on 28 bands and interpolated their outputs to 143/20 bands. As shown in Tab.4 and Fig.4 (link: https://anonymous.4open.science/r/Phy-CoSF), our continuous Phy-CoSF still consistently outperforms them.
>
> **Q2**: Spectral super-resolution evaluation
>
> **Response**: We clarify that *Table* 2 (manuscript) is the dedicated evaluation for the 20 unseen wavelengths, explicitly separating super-resolution performance from the reconstruction quality on the 143 training bands (*Table* 1).
>
> To explicitly highlight the unseen queried bands versus the training bands, we detail this division in Appendix A.2 and provide an updated spectral profile in the linked Fig. 5 (link: https://anonymous.4open.science/r/Phy-CoSF). Additionally, the lower panels of *Figure*5 and 7 (manuscript) show the comparisons for continuous spectral reconstruction (first two rows) and super-resolution(last two rows) bands.
>
> To further validate our continuous modeling, *Figure* 8 showcases visual results for arbitrary continuous queries entirely outside the standard 143/20 band sets.
>
> We sincerely apologize for the confusion and will move critical evaluations from the appendix to the main text.
>
> **Q3**: PSNR discrepancy
>
> **Response**: We respectfully clarify that this discrepancy arises from comparing two distinct datasets:
>
> 1. CAVE/KAIST Datasets: On the standard 28-band discrete benchmark, Phy-CoSF achieves 39.80 dB (*Table* 6, *Figure* 6), closely matching the 40dB reported by Qin et al. and Wu et al.
>
> 2. ICVL Dataset: Our continuous experiments used the challenging ICVL dataset, whose complex spatial-spectral structures inherently lower the PSNR baseline for all methods.
>
> We did not deliberately weaken the baselines. While our discrete performance is on par with prior works, our core contribution lies in enabling continuous spectral reconstruction and arbitrary wavelength super-resolution. Existing discrete models inherently lack these capabilities.
>
> **Q4**: Missing prior work
>
> **Response**: We thank the reviewer for highlighting these papers. We clarify that Chen et al. is already cited in our manuscript, and we will explicitly include Zhang et al. in the revision.
>
> Zhang et al. apply INRs to fully sampled HSI, bypassing the ill-posed CASSI inverse problem. Chen et al. use an end-to-end INR for CASSI without physical constraints, resulting in suboptimal accuracy and limited generalization.
>
> To avoid any ambiguity, we will refine our claim in the revision to state that Phy-CoSF is "the first physics-guided unfolding framework to jointly address continuous spectral reconstruction and super-resolution for CASSI."
>
> **Q5**: Limitations
>
> **Response**: We appreciate this insightful feedback. Inherently, INRs exhibit a spectral bias toward smooth functions. Despite mitigations from our GLAM and Mamba modules, reconstructing extremely spiky, narrow-band anomalies remains challenging and prone to over-smoothing. Consequently, applying our super-resolved spectra to precision-critical tasks (e.g., medical diagnostics) carries the risk of missing vital narrow-band details.
>
> In the revised manuscript, we will add a Limitations section to explicitly analyze these failure cases, discuss the smoothness-fidelity trade-off, and provide necessary cautions for high-stakes applications.

---

> > ### Author Rebuttal · Reviewer_A6Jp · 2026-04-02
> >
> > Thank you for the detailed response. My questions have been fully addressed.

---

> > > ### Author Response · Authors · 2026-04-03
> > >
> > > We sincerely thank the reviewer for the dedicated time and effort spent evaluating our manuscript. We are delighted that our responses have fully addressed your questions. Your constructive feedback and insightful comments have been incredibly valuable, significantly helping us to improve the clarity, rigor, and overall quality of our paper.

---

### Official Review · Reviewer_THnz · 2026-03-13

**Soundness:** 3
**Presentation:** 2
**Significance:** 2
**Originality:** 2
**Overall Recommendation:** 3
**Confidence:** 3

**Summary:**

This paper proposes Phy-CoSF. By synergizing deep unfolding networks with implicit neural representations, the authors establish a two-phase architecture—discrete-wavelength training and continuous spectral rendering—enabling high-fidelity HSI synthesis at arbitrary wavelengths. The CoSF module utilizes a triple-branch cross-domain feature mixer and a spectral synthesis head. Experiments confirm superior reconstruction fidelity and spectral detail preservation compared to state-of-the-art methods

**Compliance With Llm Reviewing Policy:**

Affirmed.

**Final Justification:**

The authors' explanation of the motivation and the core innovation of the proposed paradigm remains overly superficial, making it difficult to grasp the core ideas. Furthermore, it lacks in-depth analysis and rigorous justification. The paper lacks sufficient insight into the combination of DUN, FFT, and SSM. Although some textual explanations are provided, similar viewpoints are prevalent in many existing papers. I strongly suggest that the authors utilize illustrative diagrams to more clearly demonstrate their motivation and core innovations. This would help readers explicitly understand the targeted problem and the distinctions from previous methods; otherwise, the core contributions remain elusive. Even after carefully reading the rebuttal, I am still unable to discern the core arguments and the fundamental novelty. In essence, the proposed method appears to be merely a heuristic combination of existing techniques.

**Key Questions For Authors:**

no

**Limitations:**

see weakness

**Strengths And Weaknesses:**

Strengths：

The experimental results are promising, showing a  improvement.

Weaknesses：

1. The current manuscript lacks an in-depth analysis of the motivation, making it difficult to understand the specific limitations of existing paradigms or methods the authors aim to address. Consequently, the proposed complex network is quite confusing and appears to be a framework constructed by simply stacking various techniques, which significantly limits the novelty of the paper.

2. What is the core  novelty of the proposed method? Implicit continuous modeling has been extensively researched in the field of super-resolution, and Fourier-based frequency learning is also very common in computer vision. Furthermore, the two-stage framework mentioned by the authors has already been the subject of substantial research. Therefore, the current paper leaves the core novelty unclear. Please compare the proposed technique with existing paradigms to highlight the core novelty of the method.

3. The authors utilize both GLAM and SSM within the CDFE module; as both possess global modeling capabilities, is there any functional overlap between them? Could the structure be further simplified? Additionally, the authors need to explain the rationale for adopting a spatial-domain plus frequency-domain learning paradigm.

4. Noise Robustness Analysis: Implicit Neural Representations (INRs) are often sensitive to high-frequency noise. In practical scenarios with high noise levels, do the continuous curves generated by the SSH module exhibit severe oscillations? It is recommended to supplement the paper with experiments showing the relationship between PSNR and noise levels.

5. Table 1 indicates that the model parameters are minimal, yet the FLOPs are relatively high. It is suggested that the authors discuss in more detail the potential impact of this "lightweight parameters, heavy computation" characteristic on edge-side deployment.

6. The main body of the paper lacks ablation studies. Although they were included in the appendix, this indicates a significant issue with the organization of the manuscript. The authors must ensure that the main text covers all essential content.

---

> ### Author Rebuttal · Authors · 2026-03-31
>
> Thank you for the questions you have raised. We have provided detailed responses to each of your concerns below.
>
> **Q1**: Clarification on Motivation and Architectural Design
>
> **Response**: We apologize for the confusion regarding our motivation. Existing CASSI methods are strictly restricted to fixed, discrete spectral bands, which contradicts the physically continuous nature of hyperspectral signals. Our primary motivation is to overcome this limitation and shift the traditional discrete paradigm. By modeling the spectral volume continuously, we achieve arbitrary wavelength querying and zero-shot spectral super-resolution, perfectly aligning with physical reality for broader applications.
>
> Consequently, Phy-CoSF is not a simple stacking of techniques, but a cohesive framework meticulously designed for this specific continuous paradigm. Every module serves a mathematically and physically irreplaceable purpose:
>
> 1. DUN: It provides a mathematically rigorous optimization loop to guarantee physical data-fidelity.
>
> 2. CoSF: To break the discrete bottleneck, this module is specifically designed to replace traditional discrete priors, acting as a dynamic continuous prior.
>
> 3. CDFE: Within the CoSF, this component is uniquely required to extract a comprehensive, wavelength-agnostic latent representation.
>
> 4. SSH: It aggregates this latent representation to render spectral intensities at arbitrary continuous coordinates.
>
> **Q2**: Core novelty and comparison with existing paradigms
>
> **Response**: We thank the reviewer for the opportunity to clarify our core novelties. We do not merely borrow general concepts like INR and Fourier learning; rather, we structurally reinvent them to address the severe ill-posedness and complex physical dispersion of CASSI:
>
> 1. Paradigm Novelty: Unlike standard INRs used purely for post-processing, we are the first to embed an INR (CoSF) as a dynamic continuous prior directly within a physics-guided DUN. Our two-stage paradigm ensures that the generated continuous spectra are iteratively constrained by the physical mask and measurement at every stage.
>
> 2. Architectural Novelty: Unlike conventional CASSI networks that rely on generic U-Nets, our CoSF module utilizes a bespoke triple-branch cross-domain feature mixer to process multi-scale granularities. These are dynamically fused with an SSH to render arbitrary wavelengths.
>
> 3. Feature Novelty: Rather than applying a simple global FFT as in general Fourier learning, our CDFE innovatively cascades spatial, frequency, and channel domains. This explicitly models complex spatial-spectral interactions and reconstructs structural integrity across the continuous spectrum.
>
> **Q3**: Overlap between GLAM and SSM
>
> **Response**: We clarify that GLAM and SSM are functionally complementary, not overlapping:
>
> 1. GLAM (Spatial Domain): Specifically extracts and refines local spatial structures and high-frequency textures.
>
> 2. SSM/Fourier Mamba (Frequency Domain): Specifically captures global structural dependencies and inter-frequency relationships.
>
> 3. Rationale: Complex hyperspectral signals demand the simultaneous modeling of fine local details (best captured spatially) and broad structural correlations (best captured in frequency).
>
> 4. Validation: Our ablation study (*Table* 3, 7 and *Figure* 9 in the manuscript) demonstrates that removing either module significantly degrades performance. This confirms both are indispensable, and the structure cannot be further simplified without sacrificing reconstruction fidelity.
>
> **Q4**: Noise robustness analysis
>
> **Response**: We thank the reviewer for this constructive suggestion. To evaluate our INR's robustness, we conducted additional experiments by adding varying levels of random Gaussian noise ($\sigma = 0, 10, 20$) to the 2D measurements.
>
> As shown in Tab.3 and Fig.3 (link: https://anonymous.4open.science/r/Phy-CoSF), the generated continuous spectral curves remain remarkably smooth even under increasing noise. This demonstrates that embedding the continuous INR into our physics-guided DUN, constrained by the data-fidelity step at each stage, effectively suppresses high-frequency noise.
>
> **Q5**: Impact of "lightweight parameters, high FLOPs"
>
> **Response**: The relatively high FLOPs in Phy-CoSF stem from its multi-stage unfolding architecture and continuous coordinate querying. However, this is an efficient and necessary trade-off for high-resolution tasks. As shown in Table 5 of the manuscript, scaling traditional discrete methods to 143 bands causes their computational costs to explode quadratically (e.g., PADUT reaches 2249G FLOPs), rendering them unusable on edge devices. If accepted, we will include this discussion in the main text.
>
> **Q6**: Placement of ablation studies
>
> **Response**: We apologize for this oversight. In the revised manuscript, we will compress secondary details and move the core ablation experiments (*Table* 3 and *Figure* 9) into the main text.

---

> > ### Author Rebuttal · Reviewer_THnz · 2026-04-03
> >
> > Thank you for the rebuttal. While the authors have addressed a portion of my concerns, the core issues remain unresolved. The authors still have not provided an in-depth analysis of their motivation. The statement, "existing CASSI methods are strictly confined to fixed discrete spectral bands, contradicting the physical continuity of hyperspectral signals," makes it very difficult to understand what specific problem the authors are attempting to solve. The authors need to justify why DUN, FFT-based frequency domain modules, and SSM are specifically necessary for this field. Given that these modules have already been widely applied across various domains, the authors must provide sufficient analysis to offer novel insights; otherwise, the architecture is merely a simple stacking of existing techniques. Furthermore, the issue regarding the core innovation of the proposed paradigm remains unaddressed. Simply introducing INR into a DUN cannot be considered a fundamental paradigm shift, and the fundamental differences from existing two-stage paradigms remain unexplained.

---

> > > ### Author Response · Authors · 2026-04-04
> > >
> > > Thank you for your follow-up inquiries.
> > >
> > > **Q1**: Motivation
> > >
> > > **Response**: We appreciate the opportunity to provide a more concrete explanation. The specific problem we are solving is the spectral blind spot inherent in current discrete CASSI systems:
> > >
> > > 1. Specific Problem
> > >
> > > Current methods reconstruct a fixed, small number of discrete bands (e.g., 6 bands) from a 2D measurement, acting like a ruler with overly sparse markings. However, real-world spectral signals are continuous. If a critical spectral peak or valley occurs at 553.2 nm (e.g., indicating early-stage crop diseases in precision agriculture or subtle camouflage coatings), but the model only reconstructs channels at 540 nm and 560 nm, it misses the target. Attempting to bridge this gap with mathematical interpolation fails, interpolation merely draws a smooth line between the 540 nm and 560 nm anchors, inherently flattening the steep 553.2 nm physical peak.
> > >
> > > 2. Core Motivation
> > >
> > > Our core motivation is to fundamentally break the hardware-imposed discrete sampling limit to recover the true, continuous spectrum from a single 2D measurement. This means that when the camera captures a new scene, we no longer output fixed channels. Instead, we can zero-shot query and precisely render the spectral intensity at any arbitrary wavelength (e.g., exactly 553.2 nm) without retraining. This capability is vital for precision agriculture and industrial inspection, where resolving strict narrow-band fidelity is the absolute prerequisite for success.
> > >
> > > **Q2**: DUN, FFT, and SSM
> > >
> > > **Response**: We clarify that Phy-CoSF is not a heuristic stacking of techniques, but a targeted framework derived from reverse-engineering the physical CASSI system.
> > >
> > > 1. DUN
> > >
> > > · In CASSI, standard black-box networks (like a pure U-Net) operate blindly. They directly map the 2D input to a 3D output without any physical rules, often generating mathematical hallucinations that contradict the actual camera measurements.
> > >
> > > · DUN prevents these hallucinations by explicitly bringing the physical camera mask into the network. At every stage, it takes the current 3D prediction, simulates the physical masking process, and forces the result to match the original 2D captured image. This ensures that every generated continuous spectrum strictly obeys the real-world physical measurement constraints.
> > >
> > > 2. FFT
> > >
> > > · A core hardware component in CASSI is the prism, which physically scatters different wavelengths across the image. This mixes spatial structures and spectral colors into a highly entangled blur, which acts as a heavy spatial convolution on the data.
> > >
> > > · Unraveling this optical blur using standard spatial convolutions (like simple CNNs) is computationally difficult and ineffective. Thanks to the Convolution Theorem, this complex spatial entanglement translates to a straightforward multiplication in the frequency domain.  Thus, we employ FFT to efficiently decouple prism-induced dispersion, projecting features into the frequency domain to capture complementary global representations inaccessible to standard spatial convolutions.
> > >
> > > 3. SSM
> > >
> > > · Continuous CASSI demands stable state propagation across DUN iterations and smooth spectral continuity. While excellent at capturing non-local spatial features, Transformers tend to process spectral bands as isolated tokens rather than a smooth sequence, and lack an internal memory to bridge iterative stages.
> > >
> > > · Relying solely on Transformers fails to fully harness the iterative advantages of DUNs. In contrast, SSMs utilize a recurrent hidden-state mechanism to act as a physical memory. This allows the spectral physical states to smoothly and stably propagate and evolve from the previous unfolded stage to the next to boost performance.
> > >
> > > The role of each module is validated in our manuscript (*Table* 3, 7 and *Figure* 9)
> > >
> > > **Q3**: Paradigm
> > >
> > > **Response**: We clarify that our innovation is not a heuristic combination of INR and DUN, but a fundamental shift in the optimization paradigm.
> > >
> > > 1. Role of INR: From Post-processor to In-loop Physical Prior
> > >
> > > Unlike two-stage paradigms (like SR or NeRF) where INR acts as a data-driven post-processing renderer, our framework embeds the INR inside the hardware-constrained optimization loop. In every iteration, the INR acts as a physical prior that forces the reconstruction to strictly adhere to the CASSI system's physical laws.
> > >
> > > The INR output of each stage serves as prior knowledge for the next stage. This ensures that the continuous physical state is inherited and refined across the unrolled iterations, providing a level of stability and refinement unattainable by other two-stage paradigms.
> > >
> > > 2. Optimization: From Discrete to Continuous Functions
> > >
> > > Existing DUN methods operate within the discrete space. We pioneeringly lift the entire iterative process into a Continuous Function Space. By representing the hyperspectral volume as a continuous function throughout the reconstruction, we break the mathematical barrier of discrete sampling.

---

### Official Review · Reviewer_LZmp · 2026-03-20

**Soundness:** 2
**Presentation:** 2
**Significance:** 2
**Originality:** 2
**Overall Recommendation:** 3
**Confidence:** 5

**Summary:**

The authors leverage INR-based learning to bridge the gap between compressive sensing and continuous spectral reconstruction. The proposed method demonstrates the utility of implicit priors in enhancing the resolution and fidelity of SCI outputs.

**Compliance With Llm Reviewing Policy:**

Affirmed.

**Key Questions For Authors:**

See Weaknesses

**Limitations:**

See Weaknesses

**Strengths And Weaknesses:**

It is not clear that the proposed “continuous spectral field” provides meaningful physical benefit beyond sophisticated interpolation. The reason is that  the model is trained on discrete spectral bands, while the INR component appears to mainly approximate intermediate spectra from sampled observations, rather than recover truly new physical information beyond the sensing hardware and training data. Therefore, the continuous design just adds computational complexity and inference cost without clearly demonstrating additional scientific value over a discrete reconstruction model with interpolation.

Also, the CASSI is already with clear disadvantages:  light throughput loss, calibration difficulty, and system complexity, and there are lots of RGB to HSI/MSI methods. So it is necessary for thi s paper to  prove that CASSI measurement to continuous HSI/MSI is better than lower-cost statistical alternatives.

Finally, the paper mentioned that using DUN for interpretability, but INR with DUN is actually a learning module, without clear  interpretability. In this sense, it is hard to say how close is the generated spectral details to the true physical signals in real world.

---

> ### Author Rebuttal · Authors · 2026-03-31
>
> Thank you for your detailed review of our work. We have carefully considered your comments and offer our responses below.
>
> **Q1**: Physical Benefits of Continuous Spectral Field
>
> **Response**: We sincerely apologize for the lack of clarity regarding the physical mechanism of the continuous spectral field (CoSF). We respectfully clarify that Phy-CoSF does not perform mathematical interpolation as a post-processing step.
>
> Instead, the INR is embedded as a dynamic, continuous prior within the physics-driven optimization loop of the deep unfolding network. Crucially, when querying any unseen wavelength, the generated CoSF is strictly constrained by the physical mask $\Phi$ and the 2D measurement $y$ during the data-fidelity step of each unfolding stage. This iterative physical constraint ensures the network recovers physically meaningful information adhering to forward measurement laws, rather than calculating approximations between discrete bands.
>
> To empirically validate this, Tab.1 and Fig.1 (as shown in the link: https://anonymous.4open.science/r/Phy-CoSF) compare our continuous method against a discrete version of our model employing sophisticated interpolation. Phy-CoSF generates significantly sharper spectral details, achieving consistently higher PSNR/SSIM and lower SAM. Furthermore, our reconstructed spectral density curves align much more closely with the true physical distributions, demonstrating the scientific value and physical benefits of our continuous design over interpolation.
>
> **Q2**: Comparison between CASSI and RGB-to-HSI Methods
>
> **Response**: We thank the reviewer for this insightful point. While RGB-to-HSI is a cost-effective alternative, it is fundamentally limited by severe metamerism. RGB sensors irreversibly compress high-dimensional spectral signals into three broad channels. Consequently, RGB-to-HSI methods merely hallucinate spectra based on statistical priors, failing to recover true physical variations in complex or out-of-distribution scenarios.
>
> In contrast, CASSI captures physically encoded, multiplexed spatial-spectral measurements, offering a vastly higher physical upper bound. The disadvantages mentioned are necessary engineering trade-offs to achieve CASSI's defining advantage: high-speed, single-exposure spectral imaging without mechanical scanning or moving parts. This unique capability is highly valued in real-world applications, leading to the successful commercialization of CASSI systems for rapid industrial sorting and drone-based inspections.
>
> Furthermore, RGB spectral super-resolution targets a fundamentally different problem: mapping three discrete channels to a fixed set of target bands, whereas Phy-CoSF renders arbitrary continuous wavelengths. To empirically demonstrate CASSI's superiority, we compared our method against state-of-the-art RGB-to-HSI models (SSRNet 2024, SSRMamba 2025) under discrete reconstruction settings identical to those used in our manuscript. As shown in the linked Tab.2 and Fig.2, our measurement-based approach significantly outperforms these statistical alternatives both quantitatively and qualitatively, recovering precise spectral distributions and intensities that RGB models simply cannot retrieve. If the paper is accepted, we will include this experiment in the main text.
>
> **Q3**: Interpretability of INR within DUN and physical fidelity
>
> **Response**: We sincerely apologize for the misunderstanding. While the INR (CoSF) is indeed a data-driven module, the interpretability of our method stems from the deep unfolding network (DUN) framework itself, rather than the internal linearity of the prior.
>
> 1. Interpretability: DUNs are mathematically interpretable because they unroll optimization algorithms into explicit, alternating Data Fidelity and Prior Regularization steps. In Phy-CoSF, the CoSF simply acts as a continuous learnable prior, replacing the discrete priors (like CNNs/Transformers) used in standard DUNs. Thus, our framework inherits the exact same physics-guided interpretability.
>
> 2. Physical fidelity: CoSF operates strictly within the physics-guided alternating optimization loop, which prevents arbitrary hallucination. The generated spectra are guaranteed to align with reality because they are continuously constrained by the physical measurement and hardware mask during the Data-Fidelity step of every unfolding stage.
>
> 3. Validation: To demonstrate that the recovered details faithfully represent "true physical signals in the real world", we validate our approach using real-data experiments captured by a prototype CASSI system (as shown at the bottom of *Figure* 6 in the manuscript). If our INR+DUN were merely a hallucinating learning module lacking physical grounding, it would suffer from severe domain shift and fail on real, noisy sensor measurements. Instead, Phy-CoSF consistently recovers sharp, artifact-free spatial-spectral details on real data, significantly outperforming purely end-to-end models.

---

> > ### Author Rebuttal · Reviewer_LZmp · 2026-04-07
> >
> > I have read the rebuttal and comments from other reviewers, the conclusion is that the answer on continuous spectral field, comparison to RGB2HSI, and Interpretability of INR within DUN and physical fidelity is not convicing enough.

---

> > > ### Author Response · Authors · 2026-04-07
> > >
> > > Thank you for your continued engagement. To directly address your remaining concerns, we highlight the core physical mechanisms and objective evidence supporting our work:
> > >
> > > **Q1**: Physical Benefits of Continuous Spectral Field
> > >
> > > **Response**: Phy-CoSF is not mathematical interpolation, which inevitably flattens physical peaks. Instead, our INR is embedded within the DUN optimization loop. Every queried wavelength is iteratively constrained by the physical mask during the Data-Fidelity step. As proven by Tab. 1 and Fig. 1 (link: https://anonymous.4open.science/r/Phy-CoSF), this mask-constrained optimization recovers sharper, physically accurate details that sophisticated interpolation simply cannot achieve.
> > >
> > > **Q2**: Comparison between CASSI and RGB-to-HSI Methods
> > >
> > > **Response**: RGB-to-HSI methods suffer from severe metamerism, forcing them to rely on statistical hallucinations that fail in complex scenarios. CASSI, conversely, captures hardware-encoded physical measurements. Our new comparisons (as shown in the linked Tab. 2, Fig. 2) demonstrate that our method precisely retrieves spectral distributions that state-of-the-art RGB models (SSRNet, SSRMamba) cannot hallucinate, proving CASSI's necessity for high-fidelity reconstruction.
> > >
> > > **Q3**: Interpretability of INR within DUN and physical fidelity
> > >
> > > **Response**: Our interpretability stems from the DUN unrolling, not the INR. The INR acts solely as a learnable prior strictly confined by the physical mask and 2D measurements during the alternating Data-Fidelity steps, preventing hallucinations. This physical grounding is proven by our real-data experiments (as shown at the bottom of *Figure* 6 in the manuscript): a hallucinating black box would collapse under real sensor noise, whereas Phy-CoSF reliably recovers artifact-free, faithful details.

---

### Decision · Program_Chairs · 2026-04-30

**Decision:**

Accept (regular)

**Comment:**

This paper proposes Phy-CoSF, a physics-guided framework for snapshot compressive hyperspectral imaging that combines deep unfolding with an implicit continuous spectral representation, thereby enabling continuous spectral reconstruction and spectral super-resolution from CASSI measurements. Several reviewers consider the paper to address a meaningful and technically relevant problem, and some reviewers found the continuous reconstruction perspective to be reasonably motivated and empirically promising. The paper is also supported by extensive experimental validation, including continuous spectral reconstruction, spectral super-resolution, discrete reconstruction, and real-data experiments. Although several concerns remain after rebuttal, particularly regarding the persuasiveness of the method positioning and novelty claim, as well as the lack of deeper theoretical support for the unfolded continuous framework, the rebuttal substantially addressed the main experimental concerns, especially those related to evaluation fairness and held-out wavelength assessment. More importantly, the remaining criticisms are primarily about how the strength of the contribution should be characterized and presented, rather than pointing to clear technical flaws or invalid empirical evidence. Based on the above assessment, the AC concludes that the paper makes a technically solid and practically valuable contribution to a relatively specialized but important research direction, and therefore recommends acceptance.